# ARC-Decode: Accelerated Decoding with Risk-Bounded Acceptance

**Ying Li** [1]  **Zhaode Wang** [2]  **Zhiwen Chen** [2]  **Chengfei Lv** [2]  **Huan Wang** [1]

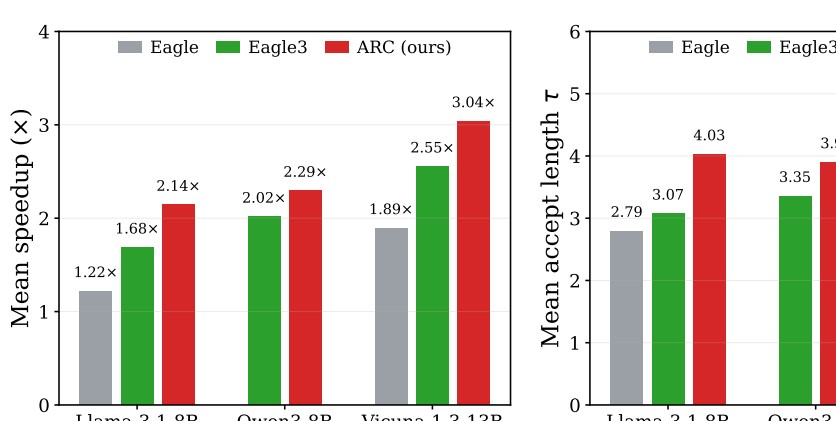

*Figure 1.* Average **speedup** (left) and average **accept length** $\tau$ (right) across tasks (means over MT-Bench, HumanEval, GSM8K, and Alpaca) for each model pair, under matched sampling settings. ARC-Decode (**ours**) attains higher speedups at longer acceptance.

## Abstract

As larger language models deliver stronger capabilities, their autoregressive inference becomes increasingly expensive. *Speculative decoding* accelerates generation by letting a fast draft propose tokens that the target model verifies in parallel. Yet under sampling ($T > 0$), observed speedups consistently lag behind those under greedy decoding, as the classical lossless verification rule tends to over-reject low-risk drafts, leading to lower acceptance rates and limited acceleration. To address this gap, we propose **ARC-Decode** (**A**cceptance with **R**isk **C**ontrol), a training-free method that augments speculative decoding without extra forward passes. ARC-Decode enables **relaxed** acceptance by identifying drafts whose acceptance is expected to induce limited local distributional deviation under a calibrated risk criterion, under a risk-controlled criterion based on Jensen–Shannon divergence. It combines confidence-based pre-verification filtering with a risk-bounded acceptance criterion derived from an analytic upper bound on the **potential** distributional deviation. Integrated into the state-of-the-

art EAGLE-3 pipeline, ARC-Decode increases accept length per cycle and reduces verification compute, achieving up to **1.6**$\times$ end-to-end speedup over EAGLE-3 under sampling with comparable generation quality across the evaluated benchmarks. Code is available at `https://github.com/NeuraLiying/ARC-Decode`.

## 1. Introduction

Modern large language models (LLMs) demonstrate strong capabilities across tasks such as search, code generation, and dialogue (Chowdhery et al., 2023; Achiam et al., 2023). These gains follow scaling trends in model size, data, and compute, with models like Qwen3-Max (Qwen, 2025) exceeding one trillion parameters. Yet inference remains bottlenecked by autoregressive next-token generation, enforcing sequential decoding and incurring high latency and cost (Shazeer, 2019). Reasoning-oriented workloads (Xu et al., 2025), such as GPT-o1, often produce longer and more complex contexts, increasing inference latency and motivating more efficient decoding. *Speculative decoding* (SD) addresses this by using a lightweight draft model to propose multiple tokens, which the target model verifies in parallel (Sun et al., 2023; Fu et al., 2024; Zhou et al., 2024; Li et al., 2024a). This transforms sequential generation into a partially parallel process, allowing a single forward pass to produce several outputs. By offloading draft generation and reducing memory-bound operations, SD lowers latency while maintaining the target model's generation behavior

[1]Westlake University [2]Alibaba Group. Correspondence to: Chengfei Lv <chengfei.lcf@taobao.com>.

*Proceedings of the 43rd International Conference on Machine Learning*, Seoul, South Korea. PMLR 306, 2026. Copyright 2026 by the author(s).

through verification. (Chen et al., 2023; Miao et al., 2024).

While SD achieves notable speedups, we observe a significant gap between greedy and sampling modes, a discrepancy absent in standard autoregressive decoding. This gap widens as the sampling temperature increases. Medusa (Cai et al., 2024) reports that higher temperatures reduce SD efficiency due to increased rejection, even when the draft and target distributions match. (Xia et al., 2024) likewise find consistent drops in acceleration as temperature rises. Across recent methods, including Speculative Sampling (Leviathan et al., 2023), EAGLE (Li et al., 2024a), HASS (Zhang et al., 2025), and EAGLE-3 (Li et al., 2025), the relative speedup under typical sampling (e.g., T = 1) can drop by over **20%**. This is concerning, as modern LLM applications typically rely on sampling-based generation for diversity and controllability.

To understand this inefficiency, we analyze the EAGLE-3 decoding pipeline and find that many rejected draft tokens are semantically equivalent to accepted ones and induce nearly identical next-step conditional distributions (Section 3.1). This suggests that lossless verification over-rejects low-risk drafts, shortening acceptance length and limiting achievable speedup. These observations raise a natural question:

*Can we safely increase draft token acceptance without compromising generation quality?*

To address this inefficiency, we propose ARC-Decode, a training-free, plug-in speculative decoding method that relaxes draft acceptance in a risk-aware manner. ARC-Decode consists of two components: (i) entropy-guided pre-verification pruning that removes low-value draft branches using a structure-preserving criterion (Section 3.2); and (ii) a risk-bounded acceptance rule that controls the induced next-step distributional divergence (Section 3.3). Both components rely only on verify-time information such as target logits, tied embeddings, and uncertainty scores, allowing seamless integration into existing speculative decoding pipelines without extra forward passes.

Applied to EAGLE-3, ARC-Decode improves acceptance length and throughput across the evaluated models and tasks. On Alpaca with LLaMA-3.1-8B, it achieves up to **1.6×** speedup under sampling, with comparable generation quality in our evaluation. Our contributions are summarized as follows.

- We introduce an **entropy-guided pruning strategy** that scores draft branches using a depth-aware confidence measure combining cumulative log-probability and target entropy, effectively filtering low-value tokens while preserving valid speculative paths.

- We propose a **risk-bounded relaxed acceptance method** that certifies next-step safety via a Lipschitz-based JS bound estimated from local logit margins

and pairwise embedding distances, and accepts tokens when the safety score exceeds a tunable threshold $\theta$.

- Experiments across multiple benchmarks and models show that our method improves decoding speed under sampling while maintaining comparable generation quality in the evaluated settings, delivering plug-and-play acceleration in the open-source speculative decoding pipeline EAGLE-3.

## 2. Related Work

**Speculative decoding.** Speculative decoding accelerates autoregressive inference by decoupling generation into a fast draft stage and a verification stage. Early approaches use either specialized draft models (Xia et al., 2023) or scaled-down target models (Liu et al., 2023; Leviathan et al., 2023), following a serial draft–then–verify pipeline (Zhang et al., 2024). Subsequent work improves draft efficiency via tree-based decoding, where multiple candidate continuations are proposed and verified using tree attention (He et al., 2024; Cai et al., 2024; Li et al., 2024a). To further enhance draft quality, later methods employ shallow draft models augmented with target hidden states or token-level guidance for multi-token prediction (Zhang et al., 2025; Li et al., 2024b). **EAGLE-3** (Li et al., 2025) extends EAGLE-2 by adopting direct token modeling with multi-layer feature fusion, leading to higher acceptance length. It has been integrated into widely used inference frameworks such as SGLang (Zheng et al., 2024) and vLLM (Kwon et al., 2023). We therefore build ARC-Decode on top of the EAGLE-3 inference pipeline and compare against its original verification policy.

**Adaptive verification and candidate selection.** Several recent works explore adaptive strategies to reduce unnecessary verification cost or improve the number of accepted tokens. DySpec (Xiong et al., 2024) introduces dynamic token-tree structures and briefly leverages KL divergence to justify draft–target closeness, while SVIP (Zhang et al., 2024) connects draft entropy to acceptance decisions during speculative generation. SpecDec++ (Huang et al., 2024) adapts candidate lengths to improve acceptance efficiency under strict verification, and Decoding Speculative Decoding (Yan et al., 2025) provides a unifying analysis of speculative decoding variants and their trade-offs. BiLD (Kim et al., 2023) uses fallback and rollback policies to dynamically decide when the large model should intervene and correct small-model predictions. MTAD (Qin et al., 2025) approximates multi-token joint decoding with an auxiliary model and verifies draft prefixes according to joint likelihood, with a multi-candidate extension for tree-structured verification. Together, these methods improve speculative decoding by adapting the draft window, candidate structure, or verification policy.

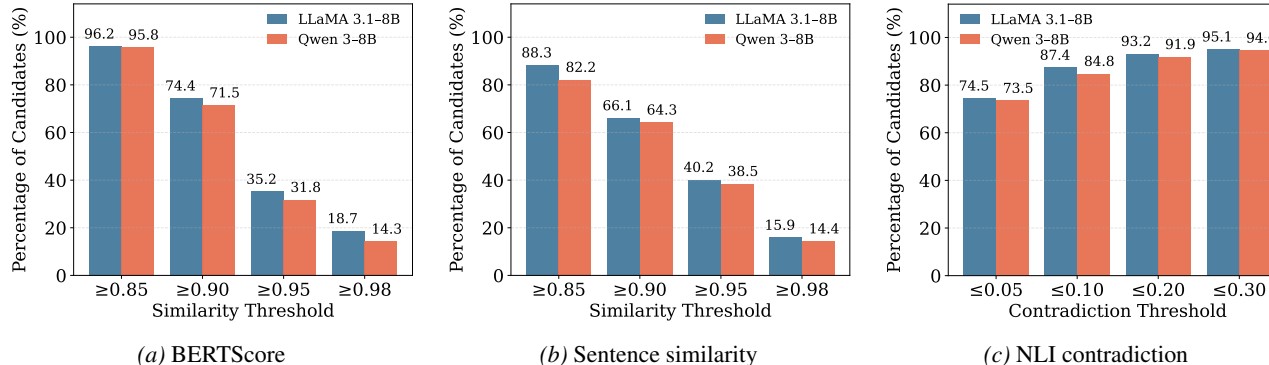

*Figure 2.* Agreement analysis on MT-Bench at temperature $T = 1$. We compare continuations seeded by rejected draft tokens against the baseline continuation seeded by the accepted token within EAGLE-3, using two backbones: Llama-3.1-8B and Qwen-3-8B. Panels: (a) BERTScore, (b) sentence similarity, (c) NLI contradiction score. The concentration of high-agreement and low-contradiction cases indicates many rejections would not materially change subsequent generation.

**Limitations of lossless verification.** Despite advances in speculative decoding, recent work questions the necessity of strict token-level verification. **MEDUSA** (Cai et al., 2024) introduces an entropy- and probability-based acceptance mechanism that avoids exact token matching with the target model. This approach improves acceptance and efficiency while preserving quality, particularly under high-temperature sampling where traditional verification yields low acceleration due to diverse outputs. **Relaxed verification** has therefore emerged as an alternative direction, allowing safe but non-identical draft tokens to be accepted when deviations are sufficiently controlled. **Judge Decoding** (Bachmann et al., 2025) observes that even with strong draft models such as GPT-4o or LLaMA-405B, accepted spans remain short under strict verification because fluent completions that only slightly diverge from the target model are frequently rejected. This exposes a key limitation of rigid token-level matching. Judge Decoding trains a compact verifier to assess token plausibility, relaxing the acceptance criterion to allow fluent but non-identical outputs. **Fuzzy Speculative Decoding** (Holsman et al., 2025) further relaxes losslessness using a divergence threshold, though it requires computing draft–target divergence at each verified position. Together, these works highlight the limitations of strict lossless verification. ARC-Decode follows this direction under sampling via compute-saving pre-verification pruning and a risk-bounded next-step acceptance rule.

## 3. ARC-Decode

This section introduces **ARC-Decode**. §3.1 analyzes speculative sampling, showing that verification dominates runtime and that many rejected drafts have negligible effect on later generation. §3.2 presents an entropy-guided pruning module, and §3.3 introduces a risk-bounded relaxed acceptance rule determining which draft tokens may be safely accepted at verification.

### 3.1. Speculative Decoding Bottlenecks under Sampling

To identify bottlenecks, we profile the EAGLE-3 speculative decoding pipeline on MT-Bench (Llama-3.1-8B). Verification dominates runtime (**70%**; Table 1), indicating that substantial compute is spent evaluating drafts later rejected by exact matching. To assess whether such rejections are often harmless, we conduct a continuation-based analysis at the *token level*. For each verification position, we force individual rejected tokens and generate 1024-token continuations under identical settings, treating each token substitution as a local perturbation within a full sequence. Semantic consistency between continuations is measured using BERTScore (Zhang et al., 2020), MPNet-base-v2 cosine similarity (Song et al., 2020), and DeBERTa-v2 NLI contradiction (He et al., 2021).

*Table 1.* Runtime breakdown on MT-Bench (Llama-3.1-8B, $T{=}1$).

| Pipeline phase | Share (%) |
| --- | --- |
| Prefill | 3.3 |
| Draft generation | 23.5 |
| **Verification forward** | **70.4** |
| Rejection sampling | 2.8 |

The resulting distributions concentrate in regions of high semantic agreement and low contradiction (Fig. 2): over 70% of rejected positions satisfy these criteria, and more than half admit at least one seemingly harmless alternative.

These results suggest that under sampling, strict lossless verification frequently discards drafts whose acceptance would not meaningfully alter downstream sequences, motivating a risk-bounded relaxed acceptance rule (§3.3). This analysis is purely diagnostic and does not inform acceptance decisions, which rely solely on next-step divergence bounds.

### 3.2. Entropy-Guided Pre-Verification Pruning

To reduce the verification overhead identified in §3.1, ARC-Decode prunes low-value branches after drafting but before calling the target model (Fig. 3(a,b)). This module operates on the already constructed draft tree and selects a compact, prefix-consistent subtree for target-side verification.

Consider a drafted tree with nodes $c \in \{0, \ldots, S-1\}$, depth

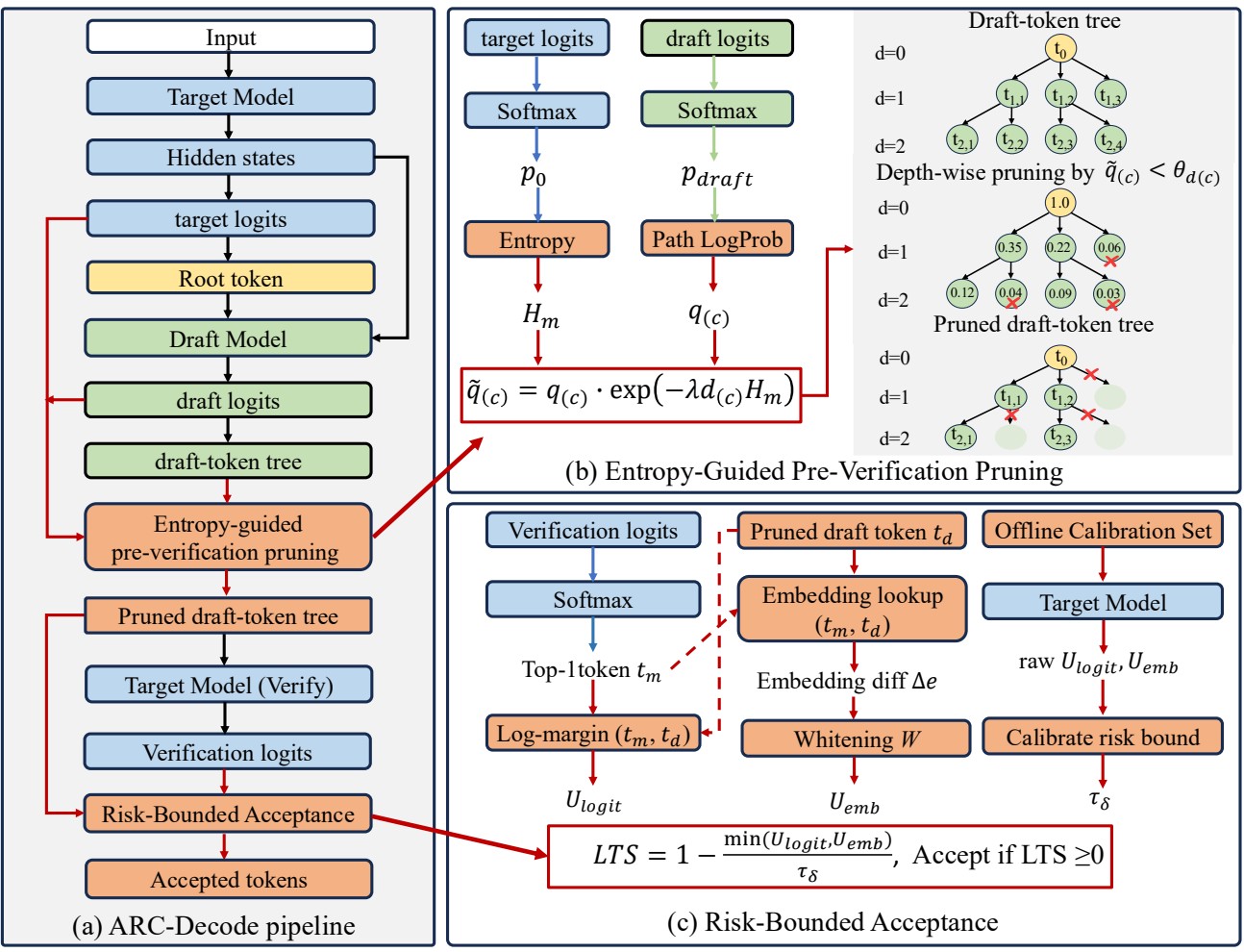

*Figure 3.* Diagram of the ARC-Decode inference pipeline for speculative sampling. (a) A speculative sampling pipeline with ARC-Decode steps highlighted (orange boxes). (b) *Entropy-Guided Pre-Verification Pruning*: combine entropy $H_m$ with path mass $q(c)$ to form $\tilde{q}(c)$, then prune using thresholded prefix-closed selection; pruned nodes are faded. (c) *Risk-Bounded Acceptance*: from verification logits and $(t_m, t_d)$, compute embedding- and logit-side bounds. Compute embedding- and logit-side local shift surrogates, combine them into an LTS score, and use the score to softly boost posterior-rejected candidates under the calibrated risk criterion.

$d(c)$, and parent pointer $\text{par}(c)$. Let $\ell(c)$ be the cumulative draft log-probability of the prefix ending at node $c$. We define the draft confidence as

$$q(c) = \exp\left(\ell(c)\right). \qquad (1)$$

To account for boundary uncertainty, let $p_b$ denote the root-selection distribution available at the speculation boundary, produced by the verification procedure. For the first cycle, it is obtained from prefill logits; in later cycles, it is obtained from the preceding verification step. When no residual correction is applied, $p_b$ coincides with the target next-token distribution. Its normalized entropy is

$$H_m = \frac{-\sum_{v \in \mathcal{V}} p_b(v) \log p_b(v)}{\log |\mathcal{V}|} \in [0, 1]. \qquad (2)$$

We then use an entropy-conditioned utility

$$\tilde{q}(c) = q(c) \exp\left(-\lambda H_m d(c)\right), \qquad \lambda \geq 0. \qquad (3)$$

When $H_m$ is small, the depth penalty is weak and the retained subtree can follow confident deep continuations; when $H_m$ is large, the penalty favors shallower verification and avoids spending budget on unreliable deep branches.

ARC-Decode selects the retained set $\mathcal{K}$ under prefix closure:

$$c \in \mathcal{K} \iff \left(\tilde{q}(c) \geq \theta\right) \wedge \left(d(c) = 0 \vee \text{par}(c) \in \mathcal{K}\right), \qquad (4)$$

where the root is always retained. We further remove weak terminal branches by a leaf threshold:

$$\text{leaf}(c) \wedge \tilde{q}(c) < \tau \implies c \notin \mathcal{K}. \qquad (5)$$

A budgeted frontier-selection variant can further use the same utility to trade depth for breadth while maintaining prefix closure. The thresholded form above primarily suppresses unreliable deep branches in high-entropy contexts.

After pruning, ARC-Decode compacts the draft tokens, tree attention mask, position indices, retrieval indices, and node confidence records before target verification. The procedure is training-free and differs from draft-length heuristics because it reshapes the verification subtree after the draft tree has already been generated.

### 3.3. Risk-Bounded Acceptance

Empirically (see §3.1), many sampling-mode rejections induce only small local shifts in the target model's next-step behavior. ARC-Decode therefore relaxes verification while keeping the standard posterior acceptance rule as a floor. If a draft token is accepted by the classical speculative sampling rule, ARC leaves this decision unchanged. If it would otherwise be rejected, ARC uses a Local Tolerance Score (LTS) to decide whether the rejection can be safely softened. Thus, the risk-control mechanism applies to the additional acceptances introduced by ARC, while the original posterior-accepted tokens follow the baseline speculative decoding path.

**Local shift surrogate.** At verification position $j$ under prefix $C$, let $x = t_d^{(j)}$ denote a drafted token, and let $t_m^{(j)} = \arg\max_t p(t \mid C)$ be the target model's top-1 anchor at the same position. The anchor is used only to measure the local effect of replacing the target-preferred continuation by the draft token. We define

$$q_{j+1} = p(\cdot \mid C, x), \qquad r_{j+1} = p(\cdot \mid C, t_m^{(j)}). \quad (6)$$

ARC measures the induced local shift by the Jensen–Shannon divergence between these two next-step distributions. In calibration and risk auditing, we compute this quantity on the union of the top $K$ supports of $q_{j+1}$ and $r_{j+1}$, followed by renormalization. We denote the resulting restricted divergence proxy by $\mathrm{JS}_K(q_{j+1}, r_{j+1})$.

**Embedding-side surrogate.** Using the target model's input embedding table, let $e_t \in \mathbb{R}^d$ denote the embedding of token $t$, and define

$$\Delta e^{(j)} = e_x - e_{t_m^{(j)}}. \quad (7)$$

The embedding displacement provides a local geometric signal for the potential next-step shift. We whiten embedding coordinates using statistics estimated from held-out calibration traces and define

$$U_{\mathrm{emb}}^{(j)} = c_e \big\| W \Delta e^{(j)} \big\|_2^2, \quad (8)$$

where $W$ is the diagonal whitening matrix and $c_e$ is a fixed calibration scale.

**Logit-side surrogate.** Let $\tilde{p}$ be the post-processed target probabilities at position $j$, after the standard logits processor

and numerical clamping. We define the target-margin signal as

$$\Delta \tilde{\ell}^{(j)} = \log \tilde{p}(t_m^{(j)}) - \log \tilde{p}(x). \quad (9)$$

A small margin indicates that the draft token is close to the target-preferred token under the current decoding distribution. The logit-side surrogate is

$$U_{\mathrm{logit}}^{(j)} = c_\ell \big( \Delta \tilde{\ell}^{(j)} \big)^2, \quad (10)$$

where $c_\ell$ is estimated from calibration traces or fixed conservatively for the decoding configuration.

**Posterior floor with LTS boost.** We combine the two signals into a local shift surrogate:

$$U^{(j)} = \min \Big\{ U_{\mathrm{emb}}^{(j)}, U_{\mathrm{logit}}^{(j)} \Big\}. \quad (11)$$

Let $\gamma$ be the surrogate threshold used for LTS gating, and define

$$s_j(x) = \mathbf{1}[\tilde{p}(x) \geq p_{\min}] \left[ 1 - \frac{U^{(j)}(C, x)}{\gamma} \right]_+, \quad (12)$$

where $[z]_+ = \max(z, 0)$. The low-probability guard $p_{\min}$ disables LTS boosting for extremely unlikely target tokens. Let

$$a_0(x) = \min \left( 1, \frac{\tilde{p}(x)}{q_{\mathrm{prop}}(x \mid C)} \right) \quad (13)$$

be the standard posterior acceptance probability, where $q_{\mathrm{prop}}$ denotes the actual proposal probability of token $x$ under the implemented draft policy. ARC keeps this posterior floor and defines

$$a_{\mathrm{ARC}}(x) = \min \big\{ 1, \, a_0(x) + \lambda_{\mathrm{LTS}} \big( 1 - a_0(x) \big) s_j(x) \big\}. \quad (14)$$

Setting $\lambda_{\mathrm{LTS}} = 0$ recovers standard speculative sampling; larger values strengthen the additional LTS-driven acceptance boost, subject to clipping at one. In this form, LTS does not replace posterior verification. It only allocates additional acceptance probability to posterior-rejected candidates whose local shift surrogate is small.

**Risk control over ARC-added acceptances.** The constants $c_e, c_\ell, \gamma, p_{\min}$, and the JS tolerance $\varepsilon_{\mathrm{JS}}$ are fixed once per backbone and decoding configuration. Calibration and held-out risk auditing are performed on traces generated under the same decoding setup. Let $E_{j,x}^+$ denote the event that token $x$ is accepted due to the ARC boost. Using the same random draw $r$ as the posterior rule, this event is $a_0(x) < r \leq a_{\mathrm{ARC}}(x)$. For eligible ARC-relaxed candidates, we target the following conditional guarantee:

$$\Pr \big[ \mathrm{JS}_K(q_{j+1}, r_{j+1}) > \varepsilon_{\mathrm{JS}} \mid E_{j,x}^+ \big] \leq \delta. \quad (15)$$

This condition states that the additional acceptances introduced by ARC remain within a calibrated top-1-anchored local shift tolerance, measured by the same $\mathrm{JS}_K$ proxy used during calibration.

**ARC-specific union bound.** Let $R_j^+$ be the set of candidates at position $j$ that survive pruning, satisfy $\tilde{p}(x) \geq p_{\min}$, and have $s_j(x) > 0$, i.e., candidates eligible for LTS-based relaxation. For each $x \in R_j^+$, let $D_{j,x}$ denote the top $K$ JS proxy between $p(\cdot \mid C, x)$ and $p(\cdot \mid C, t_m^{(j)})$, and define $\mathcal{F}_{j,x}^+ = E_{j,x}^+ \wedge \{D_{j,x} > \varepsilon_{\mathrm{JS}}\}$ as an unsafe ARC-added acceptance event. If Eq. 15 holds for each $x \in R_j^+$, then conditioning on the realized candidate set gives

$$\Pr\big[\exists x \in R_j^+ : \mathcal{F}_{j,x}^+ \mid R_j^+\big] \leq |R_j^+|\delta. \quad (16)$$

Taking expectation over the random candidate sets yields

$$\Pr\big[\exists j \leq T, \exists x \in R_j^+ : \mathcal{F}_{j,x}^+\big] \leq \delta\, \mathbb{E}\left[\sum_{j=1}^{T} |R_j^+|\right]. \quad (17)$$

This is the ARC-specific form of the sequence-level control: the risk scales with the number of relaxed candidates actually made eligible by ARC, rather than with the full vocabulary.

ARC is training-free and plug-in. It uses verify-time target logits, the target embedding table, and fixed calibration statistics, while leaving both the draft model and the target model unchanged. This complements pre-verification pruning: pruning reduces the number of verified nodes, while LTS softens low-risk posterior rejections by controlling the top-1-anchored local shift of ARC-added acceptances under the calibrated proxy criterion.

## 4. Experiments

### 4.1. Experimental Setups

**Backbones and baselines.** We base our comparisons on four target models: Llama-3.1-8B-Instruct, Qwen-3-8B, Vicuna-13B, and Llama-3.3-70B. ARC-Decode is implemented on top of the EAGLE-3 codebase and decoding pipeline (Li et al., 2025). We reuse the same draft model and verification schedule, and introduce two modifications: entropy-guided pre-verification pruning and a risk-bounded acceptance rule that replaces exact-match verification. All other decoding settings remain unchanged.

We compare ARC-Decode against EAGLE (Li et al., 2024a), HASS (Zhang et al., 2025), and EAGLE-3, as well as representative relaxed or adaptive verification baselines, including Judge Decoding (Bachmann et al., 2025), Medusa (Cai et al., 2024), Fuzzy Speculative Decoding (FSD) (Holsman et al., 2025), BiLD (Kim et al., 2023), and MTAD (Qin et al., 2025). For Medusa, we use the official pretrained Medusa-2 checkpoints released by the authors. For Judge Decoding, we re-implement the judge classifier since no pretrained judge model is publicly available. BiLD and MTAD are evaluated using EAGLE-compatible adaptations, while FSD

is evaluated as a separate inference baseline. Implementation details and hyperparameter settings for these baselines are provided in Appendix A.3. Speedup is reported relative to vanilla autoregressive decoding under matched prompts, sampling configuration, and stopping criteria.

**Tasks.** Following EAGLE and Spec-Bench (Xia et al., 2024), we evaluate four tasks using MT-Bench (Zheng et al., 2023), HumanEval (Chen et al., 2021), GSM8K (Cobbe et al., 2021), and Alpaca (Taori et al., 2023). We further include the more challenging MMLU-Pro benchmark, with results in Appendix A.3 (Table 8).

**Metrics.** We evaluate decoding using both **efficiency** and **accuracy** metrics. Efficiency includes throughput (tokens/s), i.e., wall-clock decoding speed; accepted length $\tau$, the average number of tokens accepted per verification; and speedup, the ratio over the autoregressive baseline. Accuracy is measured by four representative tasks: MT-Bench (GPT-4o-scored 1–10), HumanEval (pass@10 accuracy), GSM8K (exact match score), and Alpaca (win rate vs. GPT-4-Turbo via AlpacaEval). For HumanEval, while greedy decoding is commonly evaluated with pass@1, our sampling-based setting uses $T = 1$; therefore, we uniformly report pass@10 to reflect the multiple-sample evaluation protocol under stochastic decoding.

**Implementation and hyperparameters.** All experiments use a temperature of 1.0. For 8B–13B models, evaluation is performed on a single NVIDIA A6000 GPU, while the 70B model is evaluated on four A6000 GPUs. We expose two pruning hyperparameters: an entropy weight $\lambda = 1.0$ and a pruning threshold controlling the retained draft-tree mass. In all reported runs, the pruning configuration is fixed across tasks and backbones after calibration. A global threshold $\theta = 0.3$ is applied to the LTS score across all settings. The risk budget uses a fixed $(1-\delta) = 0.95$ quantile $\tau_\delta$, estimated on a held-out calibration set from which whitening statistics and scaling constants are also computed; all of these remain fixed during testing. Speedup is hardware-dependent, and acceptance length may vary slightly due to numerical differences. For consistency, we follow the prompt and evaluation configurations of OpenCompass (Contributors, 2023). Additional implementation details are provided in Appendix A.2.

### 4.2. Efficiency and Accuracy Results

**Efficiency.** Table 2 shows that ARC-Decode improves both acceptance length ($\tau$) and end-to-end speedup across backbones and tasks under matched decoding settings. On *Alpaca* with Llama-3.1-8B, ARC-Decode reaches **2.28×** speedup over vanilla decoding, about **1.6×** faster than EAGLE-3. Similar gains appear on Vicuna-13B, for example on *HumanEval*, where ARC-Decode improves speedup from 2.62× to 3.25×. Compared with relaxed or adaptive verification baselines, ARC-Decode generally achieves

*Table 2.* Experimental results on *MT-Bench*, *HumanEval*, *GSM8K*, and *Alpaca*. Columns report $\tau \uparrow$ (accept length) and *speedup* $\uparrow$ (end-to-end ratio vs. vanilla) for different methods. Abbrev.: L 8B = Llama-3.1-8B, Q 8B = Qwen3-8B, V 13B = Vicuna-1.3-13B, L 70B = Llama-3.3-70B. For FSD, the parameter $T$ denotes the risk threshold (not temperature).

| Model | Method | MT-bench | | HumanEval | | GSM8K | | Alpaca | |
|---|---|---|---|---|---|---|---|---|---|
| | | $\tau$ | speedup | $\tau$ | speedup | $\tau$ | speedup | $\tau$ | speedup |
| L 8B | Eagle | 3.16 | 1.15× | 3.88 | 1.46× | 2.31 | 1.17× | 1.83 | 1.09× |
| L 8B | HASS | 2.65 | 1.38× | 4.32 | 1.71× | 2.33 | 1.24× | 2.51 | 1.36× |
| L 8B | Eagle3 | 3.35 | 1.84× | 3.57 | 2.05× | 2.52 | 1.43× | 2.85 | 1.42× |
| L 8B | Judge | 4.17 | 2.01× | 3.32 | 1.97× | 2.48 | 1.24× | 3.15 | 1.74× |
| L 8B | FSD | 3.97 | 1.89× | 3.05 | 1.63× | 4.05 | 1.80× | 3.23 | 1.62× |
| L 8B | BiLD | 1.00 | 0.13× | 1.00 | 0.29× | 1.00 | 0.10× | 1.00 | 0.17× |
| L 8B | MTAD | 4.40 | 2.55× | 2.59 | 1.69× | 3.21 | 1.70× | 3.94 | 2.14× |
| L 8B | **ARC (ours)** | **5.11** | **2.61×** | **3.94** | **2.13×** | **3.71** | **1.76×** | **3.96** | **2.28×** |
| Q 8B | Eagle3 | 3.00 | 1.84× | 3.36 | 2.06× | 3.88 | 2.15× | 3.18 | 2.01× |
| Q 8B | **ARC (ours)** | **4.28** | **2.45×** | **3.42** | **2.30×** | **4.13** | **2.19×** | **3.76** | **2.23×** |
| V 13B | Eagle | 2.85 | 1.82× | 2.97 | 1.98× | 3.14 | 1.74× | 3.01 | 2.03× |
| V 13B | Eagle3 | 4.01 | 2.74× | 3.96 | 2.62× | 4.69 | 2.21× | 4.78 | 2.64× |
| V 13B | Medusa-2 | 3.26 | 2.65× | 2.67 | 2.02× | 2.78 | 2.13× | 3.24 | 2.64× |
| V 13B | FSD | 2.47 | 1.48× | 2.87 | 1.70× | 2.43 | 1.49× | 2.10 | 1.30× |
| V 13B | **ARC (ours)** | **5.28** | **3.35×** | **4.74** | **3.25×** | **5.50** | **2.52×** | **5.58** | **3.03×** |
| L 70B | FSD ($T$=0.4) | 1.52 | 1.39× | 3.43 | 2.29× | 2.48 | 1.69× | 3.17 | 2.45× |
| L 70B | FSD ($T$=0.6) | 2.54 | 1.87× | 3.56 | 2.36× | 3.21 | 1.99× | 3.91 | 2.89× |
| L 70B | FSD ($T$=0.8) | 2.37 | 1.78× | 3.56 | 2.36× | 3.66 | 2.11× | 4.14 | 3.02× |
| L 70B | EAGLE-3 | 4.04 | 3.35× | 4.11 | 3.08× | 4.37 | 2.76× | 3.82 | 3.24× |
| L 70B | **ARC (ours)** | **4.90** | **3.60×** | **4.13** | **3.23×** | **4.68** | **2.82×** | **4.20** | **3.41×** |

*Table 3.* Benchmark performance of Eagle3 and ARC-Decode on *MT-Bench*, *HumanEval*, *GSM8K*, and *Alpaca*. Metrics: *MT-Bench*: GPT-4o–judged score (1–10); *HumanEval*: pass@10 (%); *GSM8K*: EM (%); *Alpaca*: pairwise win rate (%) vs. GPT-4-Turbo (AlpacaEval).

| Model | Method | MT-bench score $\uparrow$ | HumanEval pass@10 (%) $\uparrow$ | GSM8K EM (%) $\uparrow$ | Alpaca win rate (%) $\uparrow$ |
|---|---|---|---|---|---|
| L 8B | Eagle3 | 6.77 ±0.02 | **69.9** ±1.00 | 76.6 ±0.33 | 22.8 ±0.16 |
| L 8B | **ARC (ours)** | **7.02** ±0.16 | 68.2 ±1.60 | **77.1** ±0.52 | **23.6** ±0.15 |
| Q 8B | Eagle3 | 6.83 ±0.03 | **65.8** ±1.25 | 74.2 ±0.56 | 16.4 ±0.22 |
| Q 8B | **ARC (ours)** | **6.86** ±0.07 | 64.9 ±2.01 | **74.6** ±0.59 | **16.8** ±0.17 |
| V 13B | Eagle3 | 6.07 ±0.06 | 12.0 ±0.76 | 20.3 ±0.23 | 6.63 ±0.21 |
| V 13B | **ARC (ours)** | **6.12** ±0.02 | **12.9** ±1.04 | **20.6** ±0.33 | **7.71** ±0.37 |

stronger end-to-end acceleration while maintaining longer accepted spans. For BiLD and MTAD, we report EAGLE-compatible adaptations under the shared EAGLE-3 proposal framework; their implementation details and compatibility considerations are provided in Appendix A.3.

We evaluate on Llama-3.3-70B. ARC-Decode achieves up to $3.60\times$ and higher acceptance lengths than EAGLE-3 under the same reporting protocol, indicates that ARC-Decode remains effective when scaling to larger models.

**Accuracy.** To assess the impact of ARC on generation quality, we compare ARC with EAGLE-3 on MT-Bench, HumanEval, GSM8K, and Alpaca under matched prompts, sampling hyperparameters, and stopping criteria (Table 3). We run each setting with multiple random seeds and report the mean together with the standard deviation. Across the three evaluated backbones, ARC maintains performance

comparable to EAGLE-3 on all tasks, with small fluctuations within the reported variance. In several cases, ARC yields slightly higher mean scores, such as MT-Bench on L 8B (7.02 vs. 6.77), GSM8K on L 8B (77.1% vs. 76.6%), Alpaca on Q 8B (16.8% vs. 16.4%), and all four metrics on V 13B. HumanEval shows minor decreases on L 8B and Q 8B under the $T = 1$ pass@10 protocol, but the changes are small relative to the stochastic evaluation variance. Overall, ARC maintains comparable generation quality while improving decoding efficiency, indicating that its relaxed acceptance mechanism does not introduce systematic degradation across the evaluated backbones and tasks.

### 4.3. Ablation studies

**Pruning strategies.** Table 4 compares alternative draft-tree pruning strategies under a fixed EAGLE-3 setup, including depth-based and cumulative-probability heuristics.

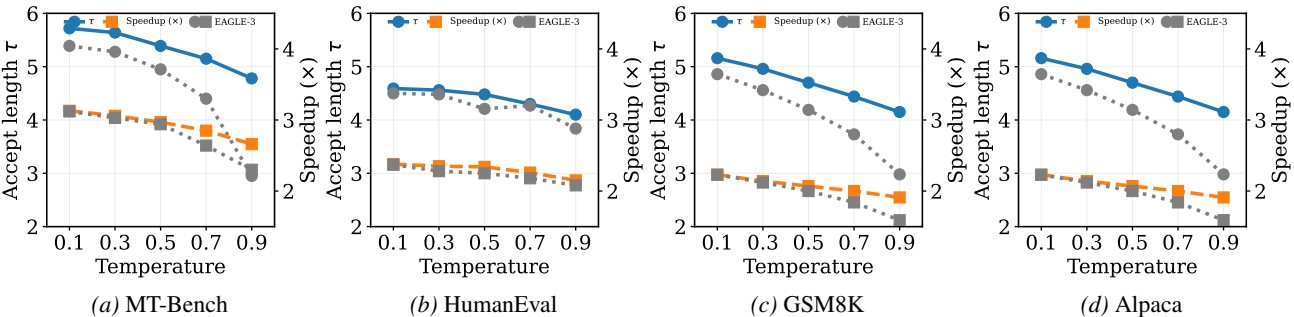

*(a)* MT-Bench      *(b)* HumanEval      *(c)* GSM8K      *(d)* Alpaca

*Figure 4.* Temperature sensitivity of ARC on Llama-3.1-8B across four tasks. We report accept length $\tau$ (blue) and speedup (orange), relative to the vanilla autoregressive baseline under temperatures $T \in 0.1, 0.3, 0.5, 0.7, 0.9$. Gray curves denote the EAGLE-3 baseline.

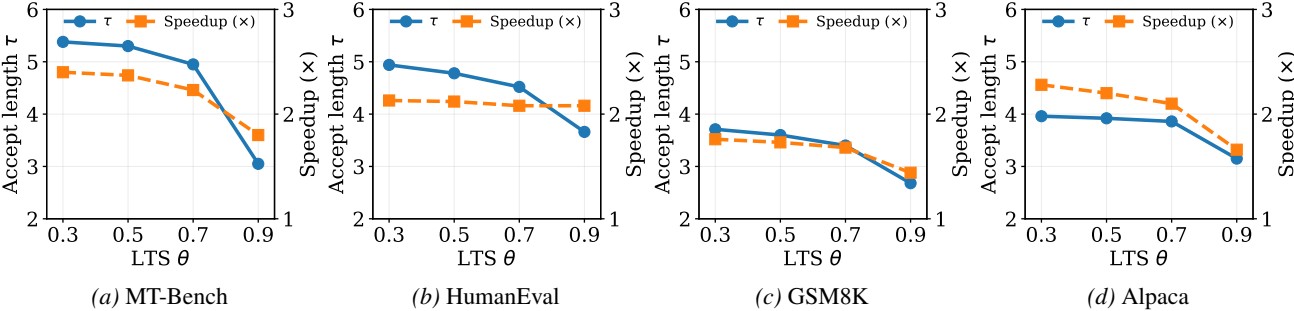

*(a)* MT-Bench      *(b)* HumanEval      *(c)* GSM8K      *(d)* Alpaca

*Figure 5.* Sensitivity to LTS threshold $\theta$ on Llama-3.1-8B across four tasks: accept length $\tau$ (blue) and speedup (orange), reported relative to a vanilla autoregressive baseline. All runs use matched prompts and stopping criteria; $\theta \in \{0.3, 0.5, 0.7, 0.9\}$.

*Table 4.* Effect of alternative pruning strategies on Qwen3-8B+EAGLE-3 decoding (MT-Bench).

| Pruning strategy | Accept ↑ | Tok/s ↑ |
|---|---|---|
| Ours (entropy-guided) | **2.70** | **56.4** |
| Depth (best $\alpha$) | 2.53 | 51.2 |
| Depth-Exp | 2.53 | 51.1 |
| Accum-Prob | 2.57 | 51.8 |

*Table 5.* Comparison between EAGLE-3 and ARC (LTS-only) across four tasks. Reported values denote relative decoding speedup over the vanilla autoregressive baseline.

| Method | MT-Bench | HumanEval | GSM8K | Alpaca |
|---|---|---|---|---|
| EAGLE-3 | 1.84× | 2.05× | 1.43× | 1.42× |
| LTS-only | 2.28× | 2.08× | 1.72× | 2.06× |

Entropy-guided pruning achieves the highest accept length and decoding throughput. Heuristic rules score nodes independently and ignore parent-closure along root-to-leaf paths, leading to fragmented draft selection, whereas our method enforces prefix closure and leaf safety, enabling more efficient use of the draft tree.

**LTS-only.** Table 5 shows that LTS alone already delivers substantial speedups over EAGLE-3 across all four tasks on Llama-3.1-8B. By selecting drafted tokens under calibrated risk bounds, LTS extends accepted prefixes and reduces verification cycles per output token, making it the dominant contributor to end-to-end acceleration.

In the full ARC-Decode framework, pruning and LTS play complementary roles: pruning removes low-confidence branches before verification, while LTS safely lengthens accepted prefixes, together maximizing decoding efficiency.

### 4.4. Sensitivity Analyses

**Temperature sensitivity.** We evaluate ARC-Decode across temperatures $T \in 0.1, 0.3, 0.5, 0.7, 0.9$ (Fig. 4). As $T$ increases, accept length $\tau$ and speedup gradually decline (e.g., MT-Bench: $\tau: 5.72 \rightarrow 4.78$, speedup: $3.13 \times \rightarrow 2.66 \times$), as higher entropy flattens the target distribution and reduces draft–target alignment. ARC-Decode consistently outperforms EAGLE-3 (gray curves) across all temperatures and tasks, with the performance gap widening at higher $T$. While both methods behave similarly at low $T$, exact-match acceptance becomes increasingly brittle as sampling variability grows. In contrast, ARC-Decode's risk-bounded criterion remains robust.

**LTS-threshold sensitivity.** We evaluate the LTS threshold $\theta$ on Llama-3.1-8B by sweeping $\theta \in 0.3, 0.5, 0.7, 0.9$ (Fig. 5). As $\theta$ increases, both accept length $\tau$ and speedup decrease, with a sharp drop beyond $\theta > 0.7$, since acceptance requires increasingly small embedding and logit differences. Since $\theta$ serves as a safety margin and we do not observe accuracy degradation in these runs, we adopt $\theta = 0.3$ by default.

Ablations (Table 5) show that LTS is the primary source of speedup, substantially increasing acceptance length without extra computation, while pruning alone yields moderate gains and further improves efficiency when combined with

*Table 6.* Paired GSM8K case studies illustrating locally accepted continuations under ARC(Ours). The first two cases show examples where LTS returns the correct answer while standard EAGLE-3 verification does not. The last case shows a complementary example where the LTS-accepted continuation is locally plausible, while the final completion differs from the gold answer.

| Relevant problem context | Local generation excerpts around the aligned difference | Final parsed answer |
|---|---|---|
| **Vacuum cleaners** Melanie sells $1/3$ of her vacuum cleaners, then 2 more, then half of the remainder, and has 5 left. | **EAGLE-3:** "Melanie **had** some vacuum cleaners left after the last house, which is 5. These 5 are half of what was left after the red house …" The completion subsequently fails to form a consistent equation for the initial number. 
 **ARC(Ours):** "Melanie **sold** a third of her vacuum cleaners at the green house. Let's call $x$ the number she started with. …After the orange house, she has $x/3 - 1$ vacuum cleaners left. …$x/3 - 1 = 5$, so $x = 18$." | EAGLE-3: 3 
 ARC(Ours): 18 
 Gold: 18 |
| **Simple interest** Brenda invests $300 at simple interest equal to $3/4$ of the original amount per year for 3 years. | **EAGLE-3:** "Simple interest **rate** means that the interest added this year is based on the original amount …After the first year …\$525 total. After the second year, the interest rate applies to the \$525 …" The completion then repeatedly applies interest to the updated balance. 
 **ARC(Ours):** "Simple interest **at** the rate of $3/4$ of the original amount per year is calculated as follows: $3/4 \times \$300 = \$225$ per year. …$\$225 \times 3 = \$675$. …$\$300 + \$675 = \$975$." | EAGLE-3: 1607.31 
 ARC(Ours): 975 
 Gold: 975 |
| **Gas cashback** Gas costs \$3.00 per gallon, with \$0.20 cashback per gallon, for a purchase of 10 gallons. | **EAGLE-3:** "The **base** price of the gas is \$3.00 per gallon. …The total base amount would be \$30.00. …They will get \$2.00 back. …$\$30 - \$2 = \$28$." 
 **ARC(Ours):** "The **cost** of gas is \$3.00 per gallon, and the person is buying 10 gallons, so the total cost is \$30. …The total cashback is \$2. …The total cost after subtracting the cashback is $\$30 - \$2 = \$8$." | EAGLE-3: 28 
 ARC(Ours): 8 
 Gold: 28 |

LTS. Sensitivity analyses over temperature and $\theta$ (Fig.4, 5) indicate that ARC-Decode degrades gracefully under stricter settings and remains robust across practical regimes. Additional details are provided in AppendixA.3.

**Calibration parameter sensitivity.** ARC-Decode relies on conservative, per-step calibration parameters to balance efficiency and robustness. The guarantees are local rather than sequence-level, as is standard in speculative decoding. Sensitivity to calibration choices (e.g., active vocabulary size $K$, smoothing $\mu$) primarily affects acceptance length: under mismatch or distribution shift, ARC-Decode becomes more conservative and smoothly reverts toward stricter verification with encouraging results.

**Paired case study of LTS soft verification.** We inspect paired GSM8K outputs from ARC-Decode and standard EAGLE-3 under the same Llama-3.1-8B decoding configuration. Since the paired traces differ at verification, this case study focuses on ARC's LTS soft verification rule. Table 6 reports the problem condition and a short generation window around the first aligned local difference. The bold token in each ARC/LTS excerpt is rejected by standard EAGLE-3 verification but accepted by LTS. These examples qualitatively illustrate ARC's local acceptance behavior and are not intended as an aggregate claim about task-level accuracy.

## 5. Conclusion

Speculative decoding parallelizes LLM inference but loses efficiency under sampling because exact verification can reject draft tokens that are locally plausible and would not

substantially alter subsequent generation. ARC-Decode addresses this problem by using a Local Tolerance Score (LTS) to identify low-risk rejected candidates and softly relax their acceptance. By increasing the number of safely accepted draft tokens, ARC-Decode extends accepted prefixes while controlling the distributional deviation introduced by relaxed acceptance, with sequence-level risk obtained by aggregating ARC-added acceptance events.

ARC-Decode integrates entropy-guided pre-verification pruning with prefix closure and LTS-based soft verification. The pruning module removes low-confidence branches while preserving extendable paths, and LTS scores posterior-rejected candidates using embedding- and logit-side evidence to support risk-controlled relaxed acceptance. Integrated into EAGLE-3, ARC-Decode significantly increases acceptance length, achieving up to $1.6\times$ additional speedup over EAGLE-3 under sampling while maintaining comparable quality across the evaluated benchmarks. Overall, ARC-Decode provides a practical plug-in acceleration mechanism for sampling-based speculative decoding with risk-controlled relaxed acceptance.

## Acknowledgements

This paper is supported by Young Scientists Fund of the National Natural Science Foundation of China (NSFC) (No. 62506305), Zhejiang Leading Innovative and Entrepreneur Team Introduction Program (No. 2024R01007), Key Research and Development Program of Zhejiang Province (No. 2025C01026), Scientific Research Project of Westlake University (No. WU2025WF003), Chinese Association for

Artificial Intelligence (CAAI) & Ant Group Research Fund - AGI Track (No. 2025CAAI-ANT-13). It is also supported by the research funds of the National Talent Program and Hangzhou Municipal Talent Program.

## Impact Statement

This paper presents work whose goal is to advance the field of Machine Learning. The proposed methods focus on algorithmic design and model optimization, without targeting any specific downstream application domain. Potential societal impacts are mainly tied to the broader deployment of large language models, where improved inference efficiency may reduce serving cost and energy consumption. We do not introduce new datasets involving human subjects or application-specific decision systems.

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

# A. Appendix

## A.1. Guarantee and Proof

**Setup and notation.** We formalize the risk controlled by ARC-Decode for the additional acceptances introduced by its LTS-based relaxation. At verification position $j$ under prefix $C$, let $x = t_d^{(j)}$ be a drafted token and let

$$t_m^{(j)} = \arg\max_t p(t \mid C)$$

be the target model's top-1 anchor at the same position. The anchor is used only to define a local reference continuation for measuring the effect of accepting $x$. We define the two next-step target distributions

$$q_{j+1}^x = p(\cdot \mid C, x), \qquad r_{j+1} = p(\cdot \mid C, t_m^{(j)}). \tag{18}$$

Following the main text, the audited discrepancy is the restricted Jensen–Shannon divergence on the union of the top-$K$ supports of $q_{j+1}^x$ and $r_{j+1}$, followed by renormalization. We denote this quantity by

$$D_{j,x} = \mathrm{JS}_K\big(q_{j+1}^x, r_{j+1}\big). \tag{19}$$

**LTS surrogate.** Using the target model's input embedding table, let $e_t \in \mathbb{R}^d$ denote the embedding of token $t$. We define

$$\Delta e_{j,x} = e_x - e_{t_m^{(j)}}. \tag{20}$$

The embedding-side surrogate is

$$U_{\mathrm{emb}}^{(j)}(x) = c_e \big\| W \Delta e_{j,x} \big\|_2^2, \tag{21}$$

where $W$ is a diagonal whitening matrix and $c_e$ is a calibration scale. Let $\tilde{p}$ denote the post-processed target probabilities at position $j$, after the same logits processor used during decoding. The logit-side surrogate is

$$U_{\mathrm{logit}}^{(j)}(x) = c_\ell \left( \log \tilde{p}(t_m^{(j)}) - \log \tilde{p}(x) \right)^2, \tag{22}$$

where $c_\ell$ is estimated on held-out traces or set conservatively for the decoding configuration. ARC combines the two as

$$U^{(j)}(x) = \min\Big\{ U_{\mathrm{emb}}^{(j)}(x), U_{\mathrm{logit}}^{(j)}(x) \Big\}. \tag{23}$$

Given the surrogate threshold $\gamma$ and a probability guard $p_{\min}$, the LTS relaxation score is

$$s_j(x) = \mathbf{1}[\tilde{p}(x) \geq p_{\min}] \left[ 1 - \frac{U^{(j)}(x)}{\gamma} \right]_+. \tag{24}$$

**Posterior floor and ARC-added acceptance event.** Let

$$a_0(x) = \min\left( 1, \frac{\tilde{p}(x)}{q_{\mathrm{prop}}(x \mid C)} \right) \tag{25}$$

be the standard posterior acceptance probability, where $q_{\mathrm{prop}}$ is the proposal probability of $x$ under the implemented draft policy. ARC keeps this posterior floor and defines

$$a_{\mathrm{ARC}}(x) = \min\{1,\ a_0(x) + \lambda_{\mathrm{LTS}}(1 - a_0(x))s_j(x)\}. \tag{26}$$

Using the same uniform random draw $r$ as the posterior rule, define the ARC-added acceptance event

$$E_{j,x}^+ = \{a_0(x) < r \leq a_{\mathrm{ARC}}(x)\}. \tag{27}$$

Thus $E_{j,x}^+$ isolates the tokens accepted due to the ARC relaxation, excluding tokens already accepted by the standard posterior rule.

**Calibration assumption.** Calibration is performed on held-out traces generated under the same backbone and decoding configuration as inference. For each eligible ARC-relaxed candidate, we target the conditional risk condition

$$\Pr\big[D_{j,x} > \varepsilon_{\mathrm{JS}} \mid E_{j,x}^+\big] \le \delta. \tag{28}$$

Here $\varepsilon_{\mathrm{JS}}$ is the audited top-$K$ JS tolerance and $\delta$ is the per-candidate risk level. This condition is empirical and calibration-based: it states that an ARC-added accepted token exceeds the local shift tolerance with probability at most $\delta$ under matched decoding conditions.

**Theorem A.1** (ARC-added local risk control). *At verification position $j$, suppose Eq. 28 holds for every candidate $x$ eligible for LTS-based relaxation. Then each ARC-added acceptance event satisfies*

$$\Pr\big[D_{j,x} > \varepsilon_{\mathrm{JS}} \mid E_{j,x}^+\big] \le \delta. \tag{29}$$

*Proof.* This follows directly from the calibration condition in Eq. 28. The event $E_{j,x}^+$ is defined to include only acceptances introduced by the ARC boost beyond the posterior floor. Therefore, the calibrated conditional risk applies exactly to the relaxed acceptances contributed by ARC. □

**ARC-specific union bound.** Let $R_j^+$ be the realized set of candidates at position $j$ that survive pruning, satisfy the probability guard $\tilde{p}(x) \ge p_{\min}$, and have $s_j(x) > 0$. For each $x \in R_j^+$, define the unsafe ARC-added acceptance event

$$\mathcal{F}_{j,x}^+ = E_{j,x}^+ \wedge \{D_{j,x} > \varepsilon_{\mathrm{JS}}\}. \tag{30}$$

Conditioning on the realized candidate set $R_j^+$, the union bound gives

$$\Pr\big[\exists x \in R_j^+ : \mathcal{F}_{j,x}^+ \mid R_j^+\big] \le \sum_{x \in R_j^+} \Pr(\mathcal{F}_{j,x}^+ \mid R_j^+) \le |R_j^+|\delta. \tag{31}$$

Taking expectation over the random candidate set gives the per-position unconditional form

$$\Pr\big[\exists x \in R_j^+ : \mathcal{F}_{j,x}^+\big] \le \delta \, \mathbb{E}[|R_j^+|]. \tag{32}$$

**Sequence-level control.** For a sequence of $T$ verification positions, applying the same union-bound argument over positions yields

$$\Pr\big[\exists j \le T, \exists x \in R_j^+ : \mathcal{F}_{j,x}^+\big] \le \delta \, \mathbb{E}\left[\sum_{j=1}^{T} |R_j^+|\right]. \tag{33}$$

If the implementation enforces a deterministic upper bound $|R_j^+| \le B$ at every verification position, Eq. 33 further implies

$$\Pr\big[\exists j \le T, \exists x \in R_j^+ : \mathcal{F}_{j,x}^+\big] \le T B \delta. \tag{34}$$

Equivalently, to target a sequence-level ARC-added risk budget $\delta_{\mathrm{seq}}$, one can set

$$\delta \le \frac{\delta_{\mathrm{seq}}}{TB}. \tag{35}$$

This is the ARC-specific sequence-level form: the accumulated risk scales with the number of candidates actually made eligible for ARC relaxation, rather than with the full vocabulary or all generated tokens.

**Connection to the surrogate construction.** The above guarantee is stated in terms of the calibrated event in Eq. 28. The role of the embedding and logit terms is to construct a practical surrogate $U^{(j)}(x)$ that identifies candidates likely to have small $D_{j,x}$. The embedding term captures local geometric proximity to the target top-1 anchor, while the logit-margin term captures closeness under the current post-processed target distribution. The constants $W, c_e, c_\ell, \gamma, p_{\min}$, and $\varepsilon_{\mathrm{JS}}$ are estimated once under the matched backbone and decoding configuration and are then fixed during inference.

**Remarks on scope.**  The risk controlled here is the top-1-anchored local next-step shift of ARC-added acceptances, measured by $\mathrm{JS}_K$. Standard posterior-accepted tokens follow the baseline speculative decoding rule and are not counted as ARC-added relaxed acceptances. The sequence-level bound aggregates these local ARC-added risk events through a union bound. It should therefore be interpreted as a distributional risk control statement for the verification decisions introduced by ARC, rather than as a deterministic guarantee of task-level correctness for every generated sequence.

*Calibration stability.* Because both surrogates depend only on the target model's embedding space and verify-time logits, which are intrinsic properties of the backbone, the calibrated constants worked across the domains and prompt styles used in our experiments. *Intuitively, LTS controls divergence in the model's own next-step distribution; as long as the backbone, vocabulary, and embedding geometry remain fixed, the underlying divergence structure remains unchanged.*

### A.2. LTS Implementation Details

**LTS Calibration and Risk Auditing.**  In ARC-Decode, $U_{\mathrm{emb}}^{(j)}$ and $U_{\mathrm{logit}}^{(j)}$ serve as calibrated local shift surrogates rather than deterministic global upper bounds. We estimate the whitening matrix $W$, the surrogate scales $c_e, c_\ell$, the LTS threshold $\gamma$, the probability guard $p_{\min}$, and the JS tolerance $\varepsilon_{\mathrm{JS}}$ on held-out traces generated under the same backbone and decoding configuration as inference.

For each calibration position, let $x$ be a drafted token and let $t_m^{(j)} = \arg\max_t p(t \mid C)$ be the target top-1 anchor. We compute the top-$K$ restricted divergence proxy $\mathrm{JS}_K(q_{j+1}, r_{j+1})$, where $q_{j+1} = p(\cdot \mid C, x)$ and $r_{j+1} = p(\cdot \mid C, t_m^{(j)})$. The two distributions are restricted to the union of their top-$K$ supports and then renormalized. The embedding-side and logit-side surrogates are computed as

$$U_{\mathrm{emb}}^{(j)} = c_e \| W(e_x - e_{t_m^{(j)}}) \|_2^2, \qquad U_{\mathrm{logit}}^{(j)} = c_\ell \big( \log \tilde{p}(t_m^{(j)}) - \log \tilde{p}(x) \big)^2.$$

We combine them as

$$U^{(j)} = \min\{U_{\mathrm{emb}}^{(j)}, U_{\mathrm{logit}}^{(j)}\}.$$

The threshold $\gamma$ controls LTS eligibility through $s_j(x) = [1 - U^{(j)}(C, x)/\gamma]_+$, together with the probability guard $\tilde{p}(x) \geq p_{\min}$. We choose the calibration constants by held-out risk auditing of ARC-added acceptances, targeting

$$\Pr\big[\mathrm{JS}_K(q_{j+1}, r_{j+1}) > \varepsilon_{\mathrm{JS}} \mid E_{j,x}^+\big] \leq \delta,$$

where $E_{j,x}^+$ denotes the event that a token is accepted due to the ARC boost rather than the posterior floor. After calibration, all constants are fixed during inference. Since $\tilde{p}$, posterior acceptance, and candidate filtering depend on the sampling setup, these constants are tied to the backbone and decoding configuration, rather than only to the downstream task.

**Calibration protocol and pseudocode.**  For each verified position $j$, calibration estimates conservative surrogates of the next-step sensitivity rather than predicting divergence exactly. Specifically, we compute: (i) the true next-step divergence $\mathrm{JS}(q_{j+1}, r_{j+1})$ over the top-$K$ union, (ii) an embedding-side raw score $U_{\mathrm{emb,raw}}^{(j)} = \| W \Delta e^{(j)} \|_2^2$ with diagonal whitening, and (iii) a logit-side raw score $U_{\mathrm{logit,raw}}^{(j)} = (\Delta \tilde{\ell}^{(j)})^2$ from post-processed probabilities. These quantities are used only to fit conservative thresholds and do not prescribe acceptance decisions at inference time, ensuring context-agnostic applicability.

**Deployment.**  At verification time we compute

$$U_{\mathrm{emb}}^{(j)} = c_s' \, \| W \Delta e^{(j)} \|_2^2, \qquad U_{\mathrm{logit}}^{(j)} = \alpha \kappa \, (\Delta \tilde{\ell}^{(j)})^2, \qquad U^{(j)} = \min\{U_{\mathrm{emb}}^{(j)}, U_{\mathrm{logit}}^{(j)}\}.$$

Then $\mathrm{LTS}^{(j)} = 1 - U^{(j)}/\tau_\delta$, and acceptance occurs when $\mathrm{LTS}^{(j)} \geq \theta$. Since all calibrated constants depend only on backbone-level geometric and probabilistic structure, the same parameters transfer across different domains without retraining. No additional forward passes are required.

### A.3. Additional Experiments

**Decoding and speculative setup.**  Unless noted otherwise, decoding uses temperature $T = 1.0$ with the unmodified EAGLE-3 verification schedule. We set the draft-tree depth to $S = 6$ and cap the number of drafted tokens per cycle at

---

**Algorithm 1** Calibration of LTS (Local Tolerance Score) parameters

---

**Require:** target model $f$, tied embeddings $E$, logits processor $g$, calibration corpus $\mathcal{D}_{\text{cal}}$, risk level $\delta$, vocab size $K$

1: Compute per-dimension inverse std from $E$ and set $W = \text{diag}(1/\hat{\sigma})$
2: Initialize lists $\mathcal{S}_{\text{JS}}, \mathcal{S}_{\text{emb}}, \mathcal{S}_{\text{logit}}$
3: **for** each sample $(C, j)$ in $\mathcal{D}_{\text{cal}}$ **do**
4:     Compute target distribution $\tilde{p} = g(f(C))$ at position $j$
5:     Let $t_m = \arg \max \tilde{p}$ and choose a candidate $t_d \neq t_m$ (e.g., uniformly among top-$K$ excluding $t_m$)
6:     Set $U_{\text{emb,raw}} = \|W(e_{t_d} - e_{t_m})\|_2^2$,    $U_{\text{logit,raw}} = (\log \tilde{p}(t_m) - \log \tilde{p}(t_d))^2$
7:     Form next-step distributions $q_{j+1} = f(C \frown t_d)$ and $r_{j+1} = f(C \frown t_m)$ restricted to the top-$K$ union of their supports
8:     Compute JS over this union
9:     Append JS to $\mathcal{S}_{\text{JS}}$, $U_{\text{emb,raw}}$ to $\mathcal{S}_{\text{emb}}$, $U_{\text{logit,raw}}$ to $\mathcal{S}_{\text{logit}}$
10: **end for**
11: $c'_s \leftarrow \text{Quantile}_{1-\delta}(\{\text{JS}/U_{\text{emb,raw}}\})$
12: $\alpha\kappa \leftarrow \text{Quantile}_{1-\delta}(\{\text{JS}/U_{\text{logit,raw}}\})$
13: For each item, define $U_{\min}^{(j)} = \min\{c'_s U_{\text{emb,raw}}^{(j)}, \alpha\kappa U_{\text{logit,raw}}^{(j)}\}$
14: $\tau_\delta \leftarrow \text{Quantile}_{1-\delta}(\{U_{\min}^{(j)}\})$
15: **return** $(W, c'_s, \alpha\kappa, \tau_\delta)$

---

`max_draft_tokens`$= 32$, with top-$k$ sampling following the backbone defaults. For Qwen-3-8B, we disable the optional "thinking" mode.

*Choice of draft length.* Our hardware environment differs from the high-throughput settings used in official EAGLE-3 evaluations: all experiments are run on NVIDIA RTX A6000 (48 GB), whose memory bandwidth and compute throughput make large draft trees significantly more latency-sensitive. As observed in prior analyses of speculative decoding efficiency (Tang et al., 2025), increasing the draft length enlarges the tree, KV-cache footprint, and attention/masking cost, and these overheads are not always amortized on memory-bound GPUs. Under this constraint, a limit of 32 drafted tokens yields more stable end-to-end latency while keeping the comparison between EAGLE-3 and ARC-Decode fair by using a fixed tree size across all methods.

**ARC-Decode settings.**    We use a global LTS threshold $\theta = 0.3$. All other ARC-Decode components follow the configurations described in the main text and Appendix, with calibration performed once and then frozen. All remaining decoding settings are kept identical to EAGLE-3.

**Calibration set.**    We calibrate all ARC-Decode constants once per backbone using a 200-prompt subset of the public OpenAssistant OASST1 dataset (English, single-turn). Prompts are uniformly sampled with a fixed seed and target lengths in [32, 256], providing a diverse collection of local decoding contexts while remaining disjoint from our evaluation domains. Calibration reuses the same decoding traces produced during standard speculative verification and introduces no additional model evaluations. All other decoding settings match those of EAGLE-3.

**Baseline implementation details.**    We provide additional implementation details for the adaptive and relaxed verification baselines used in the main experiments. For BiLD and MTAD, we implement EAGLE-compatible variants within the EAGLE-3 inference pipeline. The proposal side is fixed across methods, and only the verification or scheduling logic is changed. Specifically, all EAGLE-compatible baselines use Llama-3.1-8B-Instruct as the target model and the EAGLE3-LLaMA3.1-Instruct-8B draft checkpoint, with the shared draft-tree configuration described above.

BiLD is adapted by enabling a BiLD-style large/little scheduling rule inside the EAGLE-3 inference pipeline. We use `num_large_iters = 1`, `num_small_iters = 10`, `fallback_threshold = 0.6`, and `rollback_threshold = 5.0`. The fallback threshold controls when decoding switches from the little model to the large, and the rollback threshold controls when token-level cross-entropy under the large-model audit triggers rollback.

This baseline is an EAGLE-compatible BiLD-style adaptation rather than an optimized standalone BiLD engine. BiLD assumes an independently capable small decoder and cached large-model fallback, whereas EAGLE-3 uses a target-hidden-

*Table 7.* Empirical tightness of the JS upper bound across four benchmarks. Coverage is the fraction of accepted positions satisfying JS $\leq U_{\min}$; Wilson 95% confidence intervals are shown.

| Task | Coverage (JS $\leq U_{\min}$) | 95% CI (Wilson) |
|---|---|---|
| MT-Bench | **99.4%** | [98.7, 99.7] |
| HumanEval | **95.7%** | [93.1, 98.4] |
| GSM8K | **95.3%** | [91.7, 98.1] |
| Alpaca | **97.6%** | [96.5, 98.4] |

*Table 8.* Performance on the challenging MMLU-Pro benchmark. ARC-Decode maintains accuracy while providing higher accept length and speedup.

| Method | Accept Length | Speedup | Accuracy (%) |
|---|---|---|---|
| Vanilla | — | 1.00$\times$ | **32.4** |
| EAGLE-3 | 2.81 | 1.60$\times$ | **32.3** |
| **ARC (Ours)** | **3.50** | **1.88$\times$** | **32.3** |

state-conditioned draft module and an in-place tree-verification KV cache. This architectural mismatch makes the EAGLE draft module unsuitable as a standalone little LM and causes frequent fallback, yielding an average accepted length close to 1.0. The large-model fallback/rollback path also cannot directly reuse the standard HuggingFace-style KV cache in this shared framework. Thus, the reported BiLD numbers reflect the cost of this controlled EAGLE-compatible adaptation, not an optimized standalone BiLD serving implementation.

MTAD is adapted by replacing the verification rule inside the EAGLE-3 tree-decoding pipeline. Since the proposal side is fixed to the EAGLE-3 draft tree, we do not use a separate MTAD beam-search proposal. Instead, MTAD verifies draft paths using cumulative draft-path and target-path probabilities, following the path-level verification principle of MTAD. The method-specific acceptance threshold is set to `accept_thres = 0.5`.

Fuzzy Speculative Decoding (FSD) is evaluated as a separate inference baseline rather than being integrated into the EAGLE-3 draft-tree pipeline. We implement FSD using assisted generation with an assistant model and Jensen–Shannon divergence as the relaxed verification metric. For the Llama-3.1-8B experiments, we use Llama-3.1-8B-Instruct as the target model and Llama-3.2-1B-Instruct as the assistant model. In the main table, we report FSD under divergence thresholds $T_{\text{FSD}} \in \{0.4, 0.6, 0.8\}$, where $T_{\text{FSD}}$ denotes the FSD risk threshold and should not be confused with the sampling temperature. For the Llama-3.3-70B setting, we use Llama-3.3-70B-Instruct as the target model and Llama-3.1-8B-Instruct as the assistant model, with the same FSD-style divergence-threshold evaluation protocol.

**Empirical validation of LTS risk coverage.** To evaluate the empirical tightness of the upper bounds used by LTS, we audit accepted positions across four benchmarks (MT-Bench, HumanEval, GSM8K, Alpaca) using the Llama-3.1-8B backbone under the same decoding configuration as in §4. For each accepted draft token, we compute the true next-step JS divergence $\text{JS}(q_{t+1}, |, p_{t+1})$ and compare it against the operational bound $U_{\min} = \min U_{\text{emb}}, U_{\text{logit}}$. Across all datasets, the operational bound over-approximates the true next-step divergence for more than 95% of audited positions, with coverage consistently above 95% and reaching 99.4% on MT-Bench. These results confirm that $U_{\min}$ provides a stable and accurate surrogate for local distributional sensitivity, supporting the validity of the risk-bounded acceptance rule used by ARC-Decode.

**Evaluation on MMLU-Pro.** To assess the robustness of ARC-Decode under substantially more difficult reasoning workloads, we additionally evaluate all methods on **MMLU-Pro** (Wang et al., 2024), a large-scale benchmark containing 12,032 queries and higher task difficulty than those used in our main experiments. We compare vanilla Llama-3.1-8B, EAGLE-3, and ARC-Decode under identical decoding settings (temperature = 1.0, 5 few-shots).

Across this challenging dataset, ARC-Decode obtains the same reported accuracy as vanilla decoding and EAGLE-3 in this evaluation while achieving higher accept length and end-to-end speedup. These results indicate that ARC-Decode maintains output performance even on difficult, high-mismatch tasks.

