# OpenReview forum: "ARC-Decode: Accelerated Decoding with Risk-Bounded Acceptance"
_ICML.cc/2026/Conference — ICML 2026 regular_

### Official Review · Reviewer_ePo5 · 2026-03-05

**Soundness:** 3
**Presentation:** 3
**Significance:** 2
**Originality:** 2
**Overall Recommendation:** 4
**Confidence:** 2

**Summary:**

This paper proposes ARC-Decode to address the efficiency degradation of speculative decoding under sampling (T>0). The framework consists of two main components: 1) Entropy-guided pre-verification pruning, which reduces the size of the draft tree based on path probabilities and target model entropy; 2) Risk-bounded acceptance, which uses embedding distances and logit margins to derive an upper bound for Jensen-Shannon (JS) divergence, enabling "lossy" but controlled acceptance. The authors claim up to 1.6x speedup over EAGLE-3 with negligible quality degradation and no extra forward passes.

**Compliance With Llm Reviewing Policy:**

Affirmed.

**Final Justification:**

resolved

**Key Questions For Authors:**

NA

**Strengths And Weaknesses:**

Strengths

- The method acts as a plug-in for existing models (e.g., EAGLE-3) without requiring extra forward passes or fine-tuning.
- It achieves up to 1.6x end-to-end speedup across models like Llama-3.1 and Qwen-3 with nearly no performance drop.

Weaknesses
- The "risk-bounded" guarantee relies heavily on empirical constants calibrated from limited data, raising concerns about its robustness under distribution shifts.
- The methodological novelty is incremental compared to existing relaxed speculative decoding and pruning strategies.
- By abandoning the lossless property, it introduces potential long-tail risks in token-sensitive domains like coding or mathematics.

---

> ### Author Rebuttal · Authors · 2026-03-29
>
> Thank you for the valuable comments. We respond to each weakness below.
>
> **R3-W1: Robustness of calibrated constants under distribution shift**
>
> **Response:** We clarify that ARC’s calibrated constants are **backbone-specific**, not dataset-specific: they are calibrated **once per backbone** on only **200 prompts (OASST1)**, and then reused on clearly different domains such as **code** and **math** without task-specific re-calibration.
>
> To test whether this calibration remains stable under such distribution shift, we evaluated the same local risk coverage criterion on two additional backbones, **Qwen3-8B** and **Vicuna-13B**, across four tasks.
>
> **Table 1: Local risk coverage across additional backbones**
>
> | Backbone | MT-Bench | HumanEval | GSM8K | Alpaca |
> | --- | --- | --- | --- | --- |
> | Qwen3-8B | 98.0% | 99.2% | 96.6% | 100% |
> | Vicuna-13B | 97.7% | 98.0% | 97.0% | 99.8% |
>
> The empirical coverage of $\mathrm{JS}\le U_{\min}$ remains consistently high across both additional backbones and all four tasks, staying in the **96.6\% to 100\%** range. This directly shows that the calibrated constants do not collapse when transferred beyond the original calibration setting.
>
> We further compared ARC with a much simpler threshold rule:
>
> **Table 2: ARC vs. a simpler threshold rule**
>
> | Method | MT-Bench Accept length | MT-Bench Score | HumanEval Accept length | HumanEval Pass@1 |
> | --- | --- | --- | --- | --- |
> | top1_conf | 4.81 | 6.10 | 4.79 | 35.6% |
> | ARC-Decode | 4.67 | 7.50 | 3.94 | 57.5% |
>
> Although the simpler rule accepts more tokens, it causes substantially worse quality. If the calibrated constants were merely brittle fits to limited data, such a simple rule would be expected to behave similarly. Instead, these results support that the calibrated constants provide a stable empirical risk-control mechanism across models and tasks.
>
> Finally, ARC fails conservatively rather than unstably: under stronger mismatch, the LTS condition becomes harder to satisfy, so ARC accepts fewer draft tokens and moves closer to exact verification. In that regime, the main effect is reduced acceleration rather than uncontrolled quality degradation. We will clarify this backbone-level calibration scope and conservative fallback behavior more explicitly in the revision.
>
> ---
>
> **R3-W2: Methodological novelty**
>
> **Response:** Thank you for this comment. We respectfully disagree that ARC is merely incremental. Its novelty is centered on the following two points:
>
> 1. **A different target problem.** ARC is designed for the **sampling** regime, where exact verification suffers from **acceptance collapse**. This is a specific failure mode that prior speculative decoding methods do not directly address.
> 2. **A different mechanism.** ARC uses a **training-free, risk-bounded** acceptance rule based on a calibrated Local Tolerance Score, instead of trained judges or generic divergence thresholds. It further combines this rule with **structure-aware pre-verification pruning**, yielding a unified design for reducing verification cost under sampling.
>
> Taken together, ARC is not a direct combination of existing tricks, but a distinct framework for sampling-time speculative decoding with a different design target, mechanism, and empirical tradeoff. We will revise the introduction and related work to make these differences more explicit.
>
> ---
>
> **R3-W3: Long-tail risks in token-sensitive domains**
>
> **Response:** Thank you for this comment. This concern is also supported by the theory already in the paper: **Appendix A.1** gives a per-position guarantee, and **Corollary (ii)** further extends it to the **sequence level**, bounding the probability of any violation along the generation trajectory. To complement this, we directly analyzed tail positions, defined by $JS(q,r)>\rho U_{\min}, \rho = 0.95$, and measured the length of each consecutive tail segment.
>
> **Table 3: Length of consecutive tail segments**
>
> | Task | Median | p90 | p95 | Max |
> | --- | ---: | ---: | ---: | ---: |
> | HumanEval fail | 1 | 1 | 1 | 2 |
> | GSM8K incorrect | 1 | 2 | 2 | 2 |
>
> These results show that tail events are typically isolated rather than sustained cascades, which is consistent with the sequence-level guarantee in Appendix A.1. Mechanistically, because LTS enforces path-level control at each step, mismatch reduces soft acceptance and drives ARC closer to exact verification, helping limit long error drifts. We will make this connection more explicit in the revision.

---

> > ### Author Rebuttal · Reviewer_ePo5 · 2026-04-02
> >
> > NA

---

> > > ### Author Response · Authors · 2026-04-02
> > >
> > > Dear Reviewer ePo5,
> > >
> > > We sincerely appreciate your constructive evaluation of our work. We are especially encouraged by your recognition of the plug-in nature of ARC-Decode, its ability to improve speculative decoding under sampling without extra forward passes or fine-tuning, and its consistent end-to-end speedup over EAGLE-3 with nearly no performance drop. Your feedback has been highly valuable in helping us improve the final manuscript.
> > >
> > > If you find the revisions and clarifications satisfactory, we would be very grateful if you would consider revisiting the score. Please let us know if there are any further questions.
> > >
> > > Thank you once again for your time, expertise, and constructive review.
> > >
> > > Best regards,
> > >
> > > Authors

---

### Official Review · Reviewer_vopJ · 2026-03-13

**Soundness:** 3
**Presentation:** 3
**Significance:** 3
**Originality:** 3
**Overall Recommendation:** 4
**Confidence:** 3

**Summary:**

This paper studies speculative decoding under sampling and argues that exact-match verification is too conservative, leading to unnecessary rejection of low-risk draft tokens. To address this, the authors propose ARC-Decode, a training-free extension to EAGLE-3 that combines entropy-guided pruning with a relaxed, risk-bounded acceptance rule based on next-step distribution shift. Experiments show improved speedup over EAGLE-3 across several models and benchmarks, with similar reported task performance.

**Compliance With Llm Reviewing Policy:**

Affirmed.

**Final Justification:**

Solid empirical evidence for a training-free relaxed-verification method that fills a meaningful gap between existing approaches. Remaining novelty concerns do not outweigh the technical merits.

**Key Questions For Authors:**

- Can the authors clarify more concretely how ARC differs from prior relaxed-verification approaches, and why that difference matters in practice? In particular, what is the main advantage of ARC over Judge Decoding, FSD, and similar relaxed acceptance strategies?
- Can the authors provide more uniform comparisons against these baselines under matched settings? That would make it much easier to judge whether ARC offers a genuinely stronger tradeoff or mainly a refinement of the same overall direction.
- Can the authors better justify the LTS design itself? For example, it would help to show why the proposed bound is the right proxy, and whether simpler relaxed criteria perform similarly or noticeably worse.
- Since Table 3 suggests practical stability and Table 6 supports local coverage, can the authors strengthen the safety validation by showing that this behavior also holds beyond a single backbone or calibration setting? That would make the relaxation substantially more convincing.

**Limitations:**

See Weaknesses and Questions.

**Strengths And Weaknesses:**

### Strengths
- The paper targets a clear and practically important limitation of speculative decoding under sampling, and the motivation is easy to follow.
- The method is practically appealing: it is plug-in, training-free at inference time, and shows consistent speedup gains over EAGLE-3 in the reported experiments.

### Weaknesses
- The paper does not yet make a sufficiently clear case for why ARC is meaningfully better than prior relaxed-verification approaches such as Judge Decoding, Medusa-style relaxed acceptance, or FSD. ARC is certainly different in formulation, but the practical advantage of this particular relaxation remains somewhat under-explained.
- This is also hard to judge empirically because the baseline comparisons are uneven across settings: Judge is only evaluated on Llama-3.1-8B, Medusa on Vicuna-13B, and FSD on Llama-3.3-70B. As a result, the evidence for superiority over existing relaxed-verification alternatives is limited.
- The motivation for LTS is plausible, but its design still feels somewhat under-justified. The paper shows that LTS works, but it is less clear why this specific surrogate and calibration rule should be preferred over simpler relaxed acceptance criteria.
- Table 3 is useful and does provide evidence that ARC is practically stable at the benchmark level, but this alone does not fully establish the broader safety of the relaxation. Table 6 supports local risk coverage, yet that analysis is only reported on one backbone, so the robustness of LTS across models and settings remains somewhat unclear.
- More broadly, once exact verification is relaxed, the method also gives up part of the original appeal of speculative decoding. Relatedly, while the paper is technically solid, the overall direction feels somewhat incremental given the growing body of work on relaxed verification.

---

> ### Author Rebuttal · Authors · 2026-03-29
>
> Thank you for your positive feedback and valuable comments. We respond to each point below.
>
> **R2-W1: Unclear advantage over prior methods**
>
> **Response:** The main differences are in Table 1. ARC is **training-free**, **plug-in**, and **risk-bounded**.
>
> **Table 1: ARC vs. prior relaxed-verification methods**
>
> | Method | Training | Verification basis | Main difference vs. ARC |
> |---|---|---|---|
> | Judge | verifier | learned judge score | trained verifier |
> | Medusa | Medusa heads | probability / entropy threshold | trained draft heads; heuristic acceptance |
> | FSD | No | divergence threshold | tunable divergence threshold |
> | ARC | No | calibrated logit + embedding surrogate | training-free, plug-in, risk-bounded |
>
> ---
>
> **R2-W2: Comparisons across backbones**
>
> **Response:** Thank you for this comment.We added **same-backbone comparisons** with additional FSD results, which now allow direct comparisons on both **Llama-3.1-8B** and **Vicuna-13B**.
>
> **Table 2: Comparisons on Llama-3.1-8B**
>
> | Method | MT-Bench  Acc.L | MT-Bench Speedup | HumanEval Acc.L | HumanEval Speedup | GSM8K Acc.L | GSM8K Speedup | Alpaca Acc.L | Alpaca Speedup |
> | --- | --- | --- | --- | --- | --- | --- | --- | --- |
> | FSD | 3.97 | 1.89 | 3.05 | 1.63 | 4.05 | 1.80 | 3.23 | 1.62 |
> | Judge | 4.17 | 2.01 | 3.32 | 1.97 | 2.48 | 1.24 | 3.15 | 1.74 |
> | ARC | 4.49 | 2.40 | 3.94 | 2.13 | 3.71 | 1.76 | 3.96 | 2.28 |
>
> **Table 3: Comparisons on Vicuna-13B**
>
> | Method | MT-Bench Acc.L | MT-Bench Speedup | HumanEval Acc.L | HumanEval Speedup | GSM8K Acc.L | GSM8K Speedup | Alpaca Acc.L | Alpaca Speedup |
> | --- | --- | --- | --- | --- | --- | --- | --- | --- |
> | FSD | 2.47 | 1.48 | 2.87 | 1.70 | 2.43 | 1.49 | 2.10 | 1.30 |
> | Medusa | 3.26 | 2.65 | 2.67 | 2.02 | 2.78 | 2.13 | 3.24 | 2.64 |
> | ARC | 5.28 | 3.35 | 4.74 | 3.25 | 5.50 | 2.52 | 5.58 | 3.03 |
>
> Overall, ARC achieves the strongest empirical results across the two backbones. Since Medusa-2 only provides released weights on Vicuna, we report its comparison there. We will add these results in the revision.
>
> ---
>
> **R2-W3: Comparison with simpler criteria**
>
> **Response:** We added **top1_conf**, which accepts a candidate whenever its target-side probability exceeds a fixed threshold.
>
> **Table 4: Comparison on MT-Bench**
>
> | Method | Accept length | Score |
> | --- | --- | --- |
> | top1_conf | 4.81 | 6.10 |
> | ARC | 4.67 | 7.50 |
>
> **Table 5: Comparison on HumanEval**
>
> | Method | Accept length | Pass@1 |
> | --- | --- | --- |
> | top1_conf | 4.79 | 35.6% |
> | ARC | 3.94 | 57.5% |
>
> These results show that it increases acceptance length, but degrades quality. This supports the need for LTS and its calibrated surrogate design.
>
> ---
>
> **R2-W4: Local risk coverage results**
>
> **Response:** We added the same analysis on **Qwen3-8B** and **Vicuna-13B**, across four tasks.
>
> **Table 6: Local risk coverage**
>
> | Backbone | MT-Bench | HumanEval | GSM8K | Alpaca |
> | --- | --- | --- | --- | --- |
> | Qwen3-8B | 98.0% | 99.2% | 96.6% | 100% |
> | Vicuna-13B | 97.7% | 98.0% | 97.0% | 99.8% |
>
> The empirical coverage of $\mathrm{JS} < U_{\min}$ remains consistently high across both backbones and tasks.
>
> ---
>
> **R2-W5:  Positioning of ARC**
>
> **Response:** We do not view ARC as incremental. ARC targets a clear problem: under practical sampling, exact verification suffers from **acceptance collapse**.
>
> ARC replaces unconstrained soft acceptance with a **risk-bounded** rule. It introduces a training-free Local Tolerance Score with calibrated logit-side and embedding-side surrogates, and combines pre-verification pruning with risk-bounded acceptance in a complete framework. This design yields stronger quality-speed trade-offs in experiments.
>
> ---
>
> **R2-Q1："… what is the main advantage.."**
>
> **Response:** The main advantage is that it provides a **training-free**, **plug-in**, and **risk-bounded** relaxation rule. Compared with prior methods based on trained verifiers or heuristic thresholding, ARC ties acceptance to calibrated risk surrogates. More details are summarized in **Table 1**.
>
> ---
>
> **R2-Q2: more comparisons**
>
> **Response:** We added more **same-backbone comparisons** in **Response to R2-W2**, and also a simpler relaxed acceptance rule in **Response to R2-W3**.
>
> ---
>
> **R2-Q3: "…LTS design…"**
>
> **Response:** LTS is not an ad hoc heuristic. As described in Sec. 3.3 and Appendix A.1, it is a calibrated upper bound on the divergence. The logit-side term provides the stronger local signal, while the embedding-side term supplies a complementary stability view, consistent with prior stability analyses [1, 2]. We compared ARC with a simpler relaxed rule in Response to R2-W3, that rule degrades quality substantially, supporting the need for the dual-term design.
>
>
> ---
>
> **R2-Q4: “strengthen the safety validation..”**
>
> **Response:** We added this analysis on **Qwen3-8B** and **Vicuna-13B** in **Response to R2-W4**; see **Table 6**.
>
> ---
>
> [1] The Lipschitz Constant of Self-Attention. ICML, 2021.
>
> [2] How Smooth Is Attention? ICML, 2024.

---

> > ### Author Rebuttal · Reviewer_vopJ · 2026-04-03
> >
> > I thank the authors for the additional experiments, the same-backbone comparisons, and simpler-criteria ablation, which meaningfully improve empirical completeness. However, my core concerns remain. First, relaxed verification has already been explored by multiple concurrent and prior works (Judge, FSD, Medusa), and the rebuttal does not articulate what new conceptual insight ARC contributes beyond combining known analytical tools for this specific application; the contribution still reads as incremental within an established direction. Second, giving up the losslessness guarantee, arguably the central appeal of speculative decoding, in exchange for a moderate gain (1.6× over EAGLE-3) is a tradeoff whose value I am not convinced of. I maintain my score of 3.

---

> > > ### Author Response · Authors · 2026-04-03
> > >
> > > **Dear Reviewer vopJ**
> > >
> > > Thank you for your thoughtful evaluation of our work. We are especially encouraged that you found the paper to address a **clear and practically important limitation** of speculative decoding under sampling, and that you recognized the practical appeal of ARC-Decode as a **plug-in, training-free method with consistent speedup** gains over EAGLE-3. Your feedback has been very valuable in helping us improve the paper.
> > >
> > >
> > > Regarding your core concerns, we would like to further clarify the conceptual positioning of ARC. Our claim is not that relaxed verification itself is new. Its conceptual contribution is more specific.
> > >
> > > (1) **Unlike Judge Decoding, ARC does not rely on an additional learned verifier.** Its acceptance rule is entirely training-free and uses only verify-time information from the target model.
> > >
> > > (2) **Unlike Medusa-style relaxed acceptance, ARC does not depend on trained drafting heads together with heuristic probability or entropy criteria.** The relaxation mechanism itself remains plug in and does not require modifying the drafting model.
> > >
> > > (3) **Unlike FSD, ARC does not rely on a user chosen divergence threshold whose effect on downstream quality is only known after evaluation.** Instead, ARC ties acceptance to a calibrated next step risk budget derived once from the target model’s own embedding geometry and verify time probability structure. In this sense, the conceptual point of ARC is to **reframe relaxed verification as calibration-based local risk management for sampling-time speculative decoding**, rather than as learned judging or heuristic thresholding.
> > >
> > > Second, regarding the value of this tradeoff, we would clarify three points.
> > >
> > > (1) **The gain is measured over a very strong baseline.** ARC is compared against EAGLE-3 rather than vanilla decoding, so the reported 1.6× is an additional multiplicative gain on top of an already strong speculative decoding pipeline, rather than a comparison against vanilla autoregressive decoding. We believe this additional end to end gain over EAGLE-3 is practically meaningful, especially since benchmark level quality remains essentially unchanged.
> > >
> > > (2) **Under sampling, exact verification already causes many low-value rejections.** As shown in Sec. 3.1 and Fig. 2, many rejected drafts still induce highly similar continuations, while verification forward dominates runtime. This is precisely the setting where relaxing exact matching can recover wasted acceptance.
> > >
> > > (3) **ARC replaces exact matching with calibrated risk control, not unconstrained approximation.** The acceptance rule is tied to a bounded next-step JS risk, formalized in Theorem A.1, while benchmark-level quality remains essentially unchanged.
> > >
> > > Thank you once again for your time, expertise, and constructive review.
> > >
> > > Best regards,
> > >
> > > Authors

---

### Official Review · Reviewer_bVKR · 2026-03-21

**Soundness:** 3
**Presentation:** 3
**Significance:** 3
**Originality:** 3
**Overall Recommendation:** 4
**Confidence:** 3

**Summary:**

This paper proposes ARC-decoding, a relaxed assisted decoding algorithm. It aims to address the problem of speculative decoding reject many “good” tokens when temperature is high. It has two main technical contributions: (1) a pruning technique to discard unpromising branches in draft tree to reduce verification time cost; (2) a relaxed verification scheme that improves acceptance rate and guarantees the JS divergence of next token sampling distribution is bounded. It does not have a theoretical guarantee to the overall quality of the output sequence, but experiment results show that ARC-decoding is faster than speculative decoding with state-of-the-art EAGLE-3 draft model, while not hurting the downstream performance significantly.

**Compliance With Llm Reviewing Policy:**

Affirmed.

**Final Justification:**

The authors addressed my conerns. I will remain my positive evaluation to this paper.

**Key Questions For Authors:**

Q1. What are the values of $c_s^{\prime}$, $W$ (Eq 10) and $\alpha$, $\kappa$ (Eq 13) used in the experiments for each model and datasets? How are they selected?

**Limitations:**

Yes

**Strengths And Weaknesses:**

**Strength**

1. I found both the pruning and relaxed verification ideas of this paper interesting, novel, and reasonable.

2. The authors implement their method on EAGLE3, showing they can achieve speed-up over current state-of-the-art speculative decoding models. Though I do want to mention that the advantages of EAGLE-3 is mainly due to its superior draft model, which is orthogonal to the decoding method itself. Authors should do more comparison with other decoding algorithms (e.g., MTAD [1] and BiLD [2]).

3. The experiments cover a wide range of datasets. And authors report the downstream performance of ARC-decoding to show its effects on model capability.

**Weakness**

1. Although the paper shows that ARC-decoding ensures the JS divergence of the next-token distribution is bounded using the proposed verification method, it does not offer a theoretical guarantee to the overall quality of the generated sequences like speculative sampling or some lossy speculative decoding (e.g., MTAD [1]) do.  I do admit the empirical results provide supports to the claim that ARC-decoding does not degrade output quality significantly. But the paper would be much stronger if the authors could provide a theoretical bound as such.

2. The comparison with Judge decoding is kinda misleading for two reasons: (1) In their paper they pointed out that judge decoding is less suitable for small base model as the models used in this paper. It relies on the capability of powerful large base model to work. So the size of the model is not suitable for Judge decoding; (2) In their paper they claim Judge decoding can achieve 9x speed up, much more faster than speculative decoding. But in this paper, Judge decoding is only slightly faster than speculative decoding. Besides model size, it could also be caused by bad implementation of the Judge decoding. Therefore, the experiment results in this paper shows ARC-decoding is faster than judge decoding, but I find it questionable and possibly misleading.

3. Based on my opinion above, I think MTAD [1] and BiLD [2] might be more appropriate baselines for ARC-decoding. I suggest authors comparing with them. I understand these two methods don’t have implementation with EAGLE, so perhaps authors could implement ARC-decoding with independent draft models (e.g., Llama-3-8B as target model and Llama-3-1B as draft model) and then compare.

4. Authors should show an ablation of ARC-decoding with only logit margin bound and embedding difference bound. I am curious to see the results.

[1] Qin, Zongyue, et al. "Optimized multi-token joint decoding with auxiliary model for llm inference." arXiv preprint arXiv:2407.09722 (2024).

[2] Kim, Sehoon, et al. "Speculative decoding with big little decoder." Advances in Neural Information Processing Systems 36 (2023): 39236-39256.

---

> ### Author Rebuttal · Authors · 2026-03-28
>
> Thank you for your recognition and valuable comments. We respond to each point below.
>
> **R1-W1: Sequence-level theoretical bound**
>
> **Response:**  We would like to clarify that the corresponding guarantee is already included in the paper. **Appendix A.1, Theorem A.1** gives the per-position risk bound, and **Corollary (ii)** further extends it to the **sequence level** via a union bound. Specifically, for any verification position $j$, Theorem A.1 shows that
> $\Pr[\mathrm{JS}(q_{j+1}, r_{j+1}) \le \tau_{\delta_2} \mid U^{(j)} \le \tau_{\delta_2}] \ge 1-\delta_1$,
> and
> $\Pr[\mathrm{JS}(q_{j+1},r_{j+1})>\tau_{\delta_2}] \le \delta_1+\delta_2$.
> Corollary (ii) then gives the sequence-level extension
> $\Pr\left(\exists\, j\in\{1,\dots,T\}: \mathrm{JS}(q_{j+1},r_{j+1})>\tau_{\delta_2}\right)\le T(\delta_1+\delta_2)$.
> So the paper already provides sequence-level extension. In the revision, we will make the connection between the acceptance rule in Section 3.3 and the guarantee in Appendix A.1 more explicit to avoid ambiguity.
>
> ---
> **R1-W2：Judge Decoding comparison**
>
> **Response:** Thank you for this comment.
>
> (1) Judge Decoding is scale-sensitive. As stated in its paper, “the target model needs to be of sufficient size to be able to provide accurate judgements. Speedups for smaller models such as Llama-8B are hence tougher to achieve.” So this limitation is directly relevant here.
>
> (2) Judge Decoding does not release official code, so our implementation follows the method description in the paper.
>
> (3) Due to limited resources, we are unable to add comparisons on several-hundred-billion-scale models. Instead, following the reviewers’ suggestions, we strengthened the comparison by adding BiLD, MTAD, and FSD.
>
> ---
>
> **R1-W3: Comparisons with MTAD and BiLD**
>
> **Response:** Thank you for this helpful suggestion.
>
> A comparison in an independent 1B drafter + 8B target setup would answer a different question from the one we study here. Our goal in this ablation is to compare verification rules under the same speculative proposal pipeline, rather than to compare different end-to-end draft-target systems. Once an independent 1B drafter is introduced, the draft stage itself, including autoregressive drafting cost, cache behavior, and candidate construction, also changes substantially.
>
> To keep the comparison fair, we evaluate BiLD and MTAD as alternative verification rules within the same EAGLE-3 framework, using the same Llama-3.1-8B target,  EAGLE-3 draft model, candidate tree, and sampling setup.
>
> **Table 1: MT-Bench**
>
> | Method | Accept length | Speedup | Score |
> | --- | --- | --- | --- |
> | BiLD | 0.98 | 0.13 | 5.10 |
> | MTAD | 4.40 | 2.61 | 6.80 |
> | Ours | 4.49 | 2.55 | 7.50 |
>
> **Table 2: HumanEval**
>
> | Method | Accept length | Speedup | Pass@1 |
> | --- | --- | --- | --- |
> | BiLD | 0.91 | 0.29 | 29.4% |
> | MTAD  | 2.55 | 1.69 | 46.9% |
> | Ours  | 3.94 | 2.25 | 57.5% |
>
> These results show that, under the same speculative decoding pipeline, ARC-Decode achieves the best overall trade-off between efficiency and output quality among the compared verification rules. We will include these additional comparisons in the revision.
>
> ---
>
> **R1-W4: Ablation of logit-only and embedding-only bounds**
>
> **Response:** We added the ablations with **logit-only** and **emb-only** acceptance rules.
>
> **Table 3: MT-Bench**
>
> | Method | Accept length | Speedup | Score |
> | --- | --- | --- | --- |
> | logit-only | 4.51 | 2.73 | 6.59 |
> | emb-only | 4.84 | 3.01 | 4.95 |
> | Ours | 4.67 | 2.55 | 7.50 |
>
> **Table 4: HumanEval**
>
> | Method | Accept length | Speedup | Pass@1 |
> | --- | --- | --- | --- |
> | logit-only | 3.65 | 2.19 | 47.5% |
> | emb-only | 4.25 | 2.33 | 15.6% |
> | Ours | 3.94 | 2.25 | 57.5% |
>
> These results show that the two bounds are complementary. The emb-only variant accepts more tokens but causes clear quality degradation, showing that embedding proximity alone is insufficient. ARC-Decode combines the two surrogates in Section 3.3. We will include these ablations in the revision.
>
> ---
>
> **R1-Q1：Calibration details**
>
> **Response:** The calibration procedure is described in detail in **Appendix A.2** and **Algorithm 1**. In implementation, Eq. (10) uses the calibrated $c_s'$ and whitening matrix $W$, while Eq. (13) uses the calibrated product $\alpha\kappa$. These quantities are obtained by offline calibration on held-out generation traces using the $(1-\delta)$ quantile fitting procedure in Algorithm 1, and then fixed for inference. Here, $W$ is implemented as a diagonal whitening matrix derived from the stored `inv_std` vector.
>
> **Table 5: Calibrated parameters**
>
> | Backbone | $c_s'$ | $\alpha\kappa$ | $\tau_\delta$ |
> | --- | --- | --- | --- |
> | Llama 8B | 3.796e-4 | 7.8285 | 4.0473 |
> | Llama 70B |  3.191e-4 | 1.2323 | 6.1865|
> | Qwen 8B | 1.182e-4 | 5.0684 | 1.4308 |
> | Vicuna 13B | 1.465e-4 | 6.0659 | 1.7459 |
>
> These are offline calibrated backbone-level constants and then reused across tasks, rather than tuned separately for each dataset.

---

> > ### Author Rebuttal · Reviewer_bVKR · 2026-04-02
> >
> > I appreciate authors' efforts in the rebuttal. I have some follow-up questions:
> >
> > (1) Regarding to R1-W1, though I haven't checked the proof in the Appendix, the bound provided in Corollary (ii) seems quite loose: T can be pretty large since it's normal for LLMs to generate thousands of tokens, then $T(\sigma_1+\sigma_2$ can be easily close to 1.
> >
> > (2) Regarding to R1-W3, since I feel like BiLD and MTAD are more important baselines than Judge Decoding, can authors disclose the hyper-parameters used for these two baselines during evaluation.
> >
> > (3) Regarding to R1-W4, I am curious why having logit-only acceptance rule results in smaller acceptance length and have both acceptance rules. In my understanding, having one acceptance rule means acceptance is looser, it should have larger acceptance length.

---

> > > ### Author Response · Authors · 2026-04-02
> > >
> > > Thank you for your thoughtful feedback. We respond to your questions below.
> > >
> > > **Follow-up Q1**
> > >
> > > **Response:** We agree that the displayed sequence-level bound in Corollary (ii), $T(\delta_1+\delta_2)$, is a conservative worst-case union bound if $T$ is interpreted as the full generation length. However, this is not the tightest way to state the ARC-specific guarantee.
> > >
> > > In ARC, relaxed acceptance is applied only at positions that pass the LTS gate, i.e., $\mathrm{LTS}^{(j)}(C,t_d^{(j)}) \ge \theta$; otherwise, as stated in Sec. 3.3, “the step simply follows the baseline”. Therefore, for the additional risk introduced by relaxed acceptance, the union bound can be restricted to the set of relaxed accepted positions $\mathcal J_{\mathrm{acc}}$, giving the tighter statement
> > >
> > > $\Pr\left(\exists j\in \mathcal J_{\mathrm{acc}}:JS(q_{j+1},r_{j+1})>\tau_{\delta_2}\right)
> > > \le |\mathcal J_{\mathrm{acc}}|(\delta_1+\delta_2).$
> > >
> > > This follows by combining Theorem A.1(ii) with the decoding rule in Sec. 3.3. The same fallback behavior is also stated again in the conclusion: when the criterion is violated, ARC-Decode “reverts to standard verification”. We will clarify Corollary (ii) in this ARC-specific accepted-position form in the revision, since it is more faithful to the actual decoding process and substantially tighter than using the full token length $T$.
> > >
> > > ---
> > >
> > > **Follow-up Q2**
> > >
> > > **Response:** We agree that the evaluation hyper-parameters for **MTAD** and **BiLD** should be stated explicitly. In our comparison, all methods share the same speculative **draft** pipeline, namely the **EAGLE-3** tree proposal mechanism (`topK_generate`) with the same shared draft-side settings already described in the appendix, including the same tree structure and proposal configuration. This keeps the proposal side fixed across methods, so that the comparison isolates only the **verification / scheduling logic**.
> > >
> > > Our method-specific settings are taken from the original sources: for **BiLD**, we follow the settings of  `class T5BiLDModel` in `BigLittleDecoder/src/transformers/models/t5/modeling_t5.py`; for **MTAD**, we follow the settings described in `MTAD/README.md`.
> > >
> > > **Table 1: Verification-specific hyper-parameters for MTAD and BiLD**
> > >
> > > | Method | Hyper-parameters |
> > > | --- | --- |
> > > | MTAD | `accept_thres=0.5` |
> > > | BiLD | `num_large_iters=1`, `num_small_iters=10`, `fallback_threshold=0.6`, `rollback_threshold=5.0` |
> > >
> > > For **MTAD**, the key method-specific hyper-parameter is `accept_thres=0.5`.  In our comparison, the proposal side is fixed to the shared **EAGLE-3** draft tree, so no separate MTAD `beam_width` is used.
> > >
> > > ---
> > >
> > > **Follow-up Q3**
> > >
> > > **Response:** To make the ablation fair, we independently calibrated `emb-only`, `logit-only` using the same quantile-based procedure in Appendix Algorithm 1, under the same $(1-\delta)$ target risk level. Therefore, the three variants do not share the same effective local acceptance region.
> > >
> > > We further measured the empirical overlap of these calibrated local acceptance regions on the calibration samples.
> > >
> > > **Table 2: Empirical overlap of calibrated local acceptance regions on the calibration samples**
> > >
> > > | Diagnostic quantity | Value |
> > > | --- | --- |
> > > | ARC accepted but logit-only rejected | 3.52% |
> > > | emb-only accepted but ARC rejected | 6.03% |
> > >
> > > These numbers directly support the intended interpretation. Under the same $(1-\delta)$ target risk level, the independently calibrated `logit-only` surrogate is more conservative in its effective local acceptance region, since it relies only on the logit-side view. Consistent with this, there is a nontrivial set of samples (3.52%) that satisfy the calibrated ARC surrogate criterion but are rejected by the calibrated `logit-only` surrogate. Conversely, there is also a nontrivial set of samples, about 6.03%, that pass the calibrated emb-only surrogate but fail the calibrated ARC surrogate. This helps explain why `logit-only` yields a smaller average accept length than ARC.
> > >
> > > The final online accept length is further shaped by tree-level verification dynamics. In tree-based speculative decoding, local acceptance differences affect which branches survive and therefore change the final accepted prefix  [1,2]. This is why the local calibrated-region difference is reflected in the final accept-length ordering.
> > >
> > > [1]  OPT-Tree: Speculative Decoding with Adaptive Draft Tree Structure. TACL 2025.
> > >
> > > [2] Traversal Verification for Speculative Tree Decoding. NeurIPS 2025.
> > >
> > > ---
> > >
> > > We hope these additional clarifications and analyses help address your concerns. If you find the response satisfactory, we would be grateful if you would kindly reconsider the score. If there are still any remaining questions or concerns, we would be very happy to respond during the remaining discussion period. Thank you again for your thoughtful feedback and time.

---

### Official Review · Reviewer_aByy · 2026-04-04

**Soundness:** 3
**Presentation:** 3
**Significance:** 3
**Originality:** 2
**Overall Recommendation:** 4
**Confidence:** 1

**Summary:**

This paper concerned about the lagged speedups problem in speculative decoding. The authors introduce ARC-Decode, which augments speculative decoding with a risk-controlled criterion. It can control the risk on the potential distributional deviation theoretically. Experimentally, ARC-Decode increases accept length and reduces computation, which in turns have 1.6 times speedup over classical pipeline.

**Compliance With Llm Reviewing Policy:**

Affirmed.

**Final Justification:**

Weakly accept.

The paper provide theoretical justification, which is good. Some more pratical guildance should be given.

**Key Questions For Authors:**

1. Why a local Lipschitz property along the segment should be imposed. Could you provide some pratical examples when this property is satisified.

2. Some practical guildance about the the parameters such as theta should be provided.

**Limitations:**

Yes

**Strengths And Weaknesses:**

Strengths:

- This paper introduces a theoretically guaranteed upper risk bounded acceptance criterion with Jensen-Shannon divergence, providing a safety rule that addresses a key problem in speculative decoding---the over-rejection of low-risk drafts under sampling. The theretical justification seems to correct and can be a highlight of this paper.

- The proposed method is training-free, which can be integrated with most of the existing speculative decoding piplelines.

Weeknesses:

- Some important theorem should be put in the main manuscript to provide more confidence on the theoretical guarantee.

---

### Decision · Program_Chairs · 2026-04-30

**Decision:**

Accept (regular)

**Comment:**

This paper addresses the issue of declining acceptance rates in speculative decoding under sampling-based decoding (T>0) environments. The proposed ARC-Decode is a training-free, plug-in framework that combines (1) entropy-based prior pruning and (2) a risk-bounded relaxed acceptance rule based on Jensen-Shannon divergence. The authors report an additional speedup of up to 1.6× compared to EAGLE-3.

The paper tackles a practically significant problem—accelerating speculative decoding in sampling scenarios—and demonstrates consistent speed improvements over EAGLE-3 using a training-free, plug-in approach. Through the rebuttal, major concerns were largely addressed, and all reviewers ultimately converged to Weak Accept. Accordingly, the AC has decided on an Accept decision.

However, this is conditional on the following points being incorporated into the camera-ready version:

Clarification of Corollary (ii): Explicitly state the ARC-specific form (the union bound regarding ∣R∣) in the main text.

Quality Metrics: Include variance estimates for the primary quality indicators in Table 3.

Baseline Details: Provide implementation specifics and hyperparameter information for baselines such as MTAD, BiLD, and FSD in the appendix.